

# Power of parametric and non-parametric tests for trend detection in annual maximum series

Vincenzo Totaro, Andrea Gioia, Vito Iacobellis

Department of Civil, Environmental, Land, Building Engineering and Chemistry (DICATECh), Polytechnic University of
Bari, Bari, 70125, Italy

*Correspondence to*: Vincenzo Totaro (vincenzo.totaro@poliba.it)

**Abstract.** The need of fitting time series characterized by the presence of trend or change points has generated in latest years an increased interest in the investigation of non-stationary probability distributions. Considering that the available hydrological time series can be recognized as the observable part of a stochastic process with a definite probability distribution, two main
topics can be tackled in this context: the first one is related to the definition of an objective criterion for choosing whether the stationary hypothesis can be adopted, while the second one regards the effects of non-stationarity on the estimation of distribution parameters and quantiles for assigned return period and flood risk evaluation. Although the time series trend or change points can be recognized using classical tests available in literature (e.g. Mann-Kendal or CUSUM test), for design purpose it is still required the correct selection of the stationary or non-stationary probability distribution. By this light, the
focus is shifted toward model selection criteria which implies the use of parametric methods with all related issues on parameters estimation. The aim of this study is to compare the performance of parametric and non-parametric methods for trend detection analysing their power and focusing on the use of traditional model selection tools (e.g. Akaike Information Criterion and Likelihood Ratio test) within this context. Power and efficiency of parameter estimation, including the trend coefficient, were investigated through Monte Carlo simulations using Generalized Extreme Value distribution as parent with
selected parameter sets.

## 1 Introduction

Long and medium-term prediction of extreme hydrological events under non-stationary conditions, is one of the major challenges of our times. Streamflow, as well as temporal rainfall and many other hydrological phenomena, can be considered stochastic processes (Chow, 1964), i.e. families of random variables with an assigned probability distribution (Koutsoyiannis
and Montanari, 2015), while time series are the observable part of this process. One of the main goals of the frequency analysis of extreme events is the estimation of distribution quantiles related to a certain non-exceedance probability. They are usually obtained after fitting a probabilistic model to observed data. *Quae cum ita sint*, detecting the existence of time-dependence in a stochastic process, has to be considered a necessary task in the statistical analysis of recorded time series.



According to Salas (1993) "a hydrological time series is stationary if is free of trends, shifts, or periodicity (cyclicity)". Starting from this statement several considerations can be done in updating some important hydrological concepts while assuming that non-exceedance probability varies with times or other covariates. For example, return period may be reformulated in two different ways, Expected Waiting Time (EWT, Olsen et al., 1998) or Expected Number of Events (ENE, Parey et al., 2007, 2010) which lead to a different evaluation of quantiles into a non-stationary approach. As proved by Cooley (2001), EWT and ENE are differently affected by non-stationarity, possibly producing ambiguity in engineering design practice (Du et al., 2015; Read and Vogel, 2015). Salas and Obeysekera (2014) provided a detailed report about relationships between stationary and non-stationary EWT values within a parametric approach for the assessment of non-stationary conditions. In such a framework, a strong relevance is given to statistical tools for detecting changes in non-normally distributed time series (Kundewicz and Robson, 2004).

On the other hand, the vast majority of research undertaken about climate change and detection of non-stationary conditions has been so far developed through non-parametric approaches. One of the most used non-parametric measures of trend is the Sen's slope (Gocic and Trajkovic, 2013). Also, a wide gamma of non-parametric tests for detecting non-stationarity in time series is available (e.g. Kundewicz and Robson, 2004 ). Statistical tests include Mann-Kendall (*MK;* Mann, 1945; Kendall, 1975) and Spearman (Lehmann, 1975) for detecting trends, Pettitt (Pettitt, 1978) and CUSUM (Smadi and Zghoul, 2006) for change point detection. All of these tests are based on a specific null hypothesis and have to be performed for an assigned significance level. Non-parametric tests are usually preferred to parametric ones because they are distribution-free and do not require knowledge of the parent distribution. In the frequency analysis of extreme events they are also suggested being less sensitive to the presence of outliers with respect to parametric tests (Wang et al., 2005).

In general, the use of statistical tests involves different errors, such as type I (rejecting the null hypothesis when it is true) and type II (accepting the null hypothesis when it is false). An important characteristic is the power of such tests, i.e. the probability of rejecting the null hypothesis when it is false. It is worth noting that, as already proven by numerical experiments by Yue et al. (2002a), the power of the Mann-Kendall test, despite its non-parametric structure, actually shows a strong dependence on the type and parametrization of the parent distribution.

In a parametric approach, the estimation of quantiles of the extreme events distribution requires the search for the underlying distribution and for time-dependant hydrological variables, providing the identification of a model, which can be stationary or not (Montanari and Koutsoyiannis, 2014). In other words, it is necessary to define if variables are iid (independent identically distributed) or i/nid (independent/non identically distributed) and accordingly to select between a stationary or not-stationary distribution model (Serinaldi and Kirsby, 2015).

In this perspective, the detection of non-stationarity may exploit, besides traditional statistical tests, well known properties of model selection tools. Even in this case several measures and criteria are available for selecting a best-fit model, among these we find Akaike Information Criterion (AIC, Akaike, 1974), Bayesian Information Criterion (BIC, Schwarz, 1978) and Likelihood Ratio test (LR, Coles, 2001), the latter is suitable when dealing with nested models.





The purpose of this paper is to provide further insights on the use of parametric and non-parametric approaches in the framework of frequency analysis of extreme events under non-stationary conditions. The comparison between those different approaches is not straightforward. Non-parametric tests do not require the knowledge of the parent distribution and their

properties strongly rely on the choice of a null hypothesis. Parametric methods for model selection, on the other hand, require the selection of the parent distribution and the estimation of its parameters but are not necessarily associated with a specific null hypothesis. Nevertheless, in both cases the evaluation of the rejection threshold is usually based on a statistic measure of trend that, under the null hypothesis of stationarity, follows a specific distribution (e.g. gaussianity of the Kendall statistic for the MK non-parametric test; $\chi^2$ distribution of deviance statistic for the LR parametric test).

Considering pros and cons of different approaches, we believe that specific remarks should be made about the use of parametric or non-parametric methods for the analysis of extreme event series. For this purpose, we set up a numerical experiment to compare performances of: 1 the MK as a non-parametric test for trend detection, 2 the LR parametric test for model selection, 3 the $AIC_R$ parametric test as defined in section 2.4. In particular, the $AIC_R$ is a measure for model selection, based on the AIC, whose distribution was numerically evaluated, under the null hypothesis of a stationary process, for comparison purposes with

other tests.

We aim to provide (i) a comparison of test power between MK, LR and $AIC_R$, (ii) a sensitivity analysis of test power to parameters of a known parent distribution used to generate sample data, (iii) an analysis of the influence of sample size on test power and significance level.

We conducted the analysis using Monte Carlo techniques, by generating samples from parent populations assuming one of the

most popular extreme event distributions, the Generalized Extreme Value (Jenkinson, 1955), with linear (and without any) trend in the position parameter. From generated samples we numerically evaluated the power and significance level of tests for trend detection, using MK, LR and $AIC_R$. For the latter we also checked the option of using the modified version $AIC_c$, suggested by Sugiura (1978) for smaller samples.

Considering that parametric methods involve the estimation of the parent distribution parameters, we also analysed the

efficiency of the Maximum Likelihood (ML) estimator used in trend assessment by comparing the sample variability of the ML estimate of trend with the non-parametric Sen's slope. We also scoped the sample variability of GEV parameters in the stationary and non-stationary cases.

## 2 Methodological framework

This section is divided into five parts Subsections 2.1, 2.2 and 2.3 report main characteristics of respectively, *MK*, *LR* and

$AIC_R$ based test. In the fourth subsection the probabilistic model used for generations, based on the use of the GEV distribution, is described in the stationary and non-stationary cases. Subsection 2.5 outlines the procedure for numerical evaluation of tests' power and significance level.



## 2.1 The Mann-Kendall test

Hydrological time series are often composed by non-normally independent realizations of phenomena, and this characteristic makes the use of non-parametric trend tests very attractive (Kundzewicz and Robson, 2004). Mann-Kendall test is a widely used rank-based tool for detecting monotonic, and not necessarily linear, trends. Given a random variable z, and assigned a sample of $L$ independent data $\mathbf{z} = (z_1, \dots, z_L)$, the Kendall S statistic (Kendall, 1975) can be defined as:

$$S = \sum_{i=1}^{L-1} \sum_{j=i+1}^{L} sgn(z_j - z_i), \tag{1}$$

with *sgn* sign function.

The null hypothesis of this test is the absence of any statistically significant trend in the sample, while it is contemplated by the alternative hypothesis. Yilmaz and Perera (2014) reported that serial dependence can lead to a more frequent rejecting of null hypothesis. For $L \geq 8$, Mann (1945) reported how Eq. (1) is an approximatively normally distributed variable with zero mean and variance that, in the presence of $t_m$ $m$-length ties, can be expressed as:

$$V = \frac{L(L-1)(2L+5) - \sum_{m=1}^{n} t_m m(m-1)(2m+5)}{18}.$$

In practice, Mann-Kendall test is performed using the $Z$ statistic

$$Z = \begin{cases} \frac{S-1}{\sqrt{V(S)}} & S > 0 \\ 0 & S = 0 \\ \frac{S+1}{\sqrt{V(S)}} & S < 0 \end{cases},$$

which follows a standard normal distribution. With this approach, it is simple to evaluate p-value and compare it with an assigned level of significance or, equivalently, to evaluate the $Z_\alpha$ threshold value to be compared with $Z$, where $Z_\alpha$ is the $(1 - \alpha)$ quantile of a standard normal distribution.

Yue et al. (2002b) observed that autocorrelation in time series can influence the ability of *MK* test in detecting trends. For avoiding this problem, a correct approach in trend analysis should contemplate a preliminary check for autocorrelation and, if necessary, the application of pre-whitening procedures.

A non-parametric tool for a reliable estimation of a trend in a time series with $N$ pairs of data is the Sen's slope estimator (Sen, 1968):

$$\delta_j = \frac{z_i - z_k}{i - k}, \qquad j = 1, \dots, N \tag{2}$$

being $j > k$. Sorting in ascending order the $\delta_j$'s, Sen's slope estimator can be defined as their median $\delta$.





## 2.2 Likelihood Ratio Test

The Likelihood Ratio statistical test allows to compare two candidate models. Like its name suggests, it is based on the evaluation of the likelihood function of different models.

The LR test has been used different times (Tramblay et al., 2013; Cheng et al., 2014; Yilmaz et al., 2014) for selecting between stationary and non-stationary models in the context of nested models. Given a stationary model characterized by a parameter set $\boldsymbol{\theta}_{st}$ and a non-stationary model, with parameter set $\boldsymbol{\theta}_{ns}$, if $\ell(\widehat{\boldsymbol{\theta}}_{st})$ and $\ell(\widehat{\boldsymbol{\theta}}_{ns})$ are their respective maximized log-likelihoods, the Likelihood Ratio test can be defined through the deviance statistic

$$D = 2[\ell(\widehat{\boldsymbol{\theta}}_{ns}) - \ell(\widehat{\boldsymbol{\theta}}_{st})], \tag{3}$$

D is approximately, for large L, $\chi^2_m$ distributed, with $m = dim(\boldsymbol{\theta}_{ns}) - dim(\boldsymbol{\theta}_{st})$ degrees of freedom. The null hypothesis of stationarity is rejected if $D > C_\alpha$, where $C_\alpha$ is the $(1 - \alpha)$ quantile of $\chi^2_m$ distribution (Coles, 2001).

Besides the analysis of power, we also checked (in subsection 3.3) the approximation $D \sim \chi^2_m$ as a function of the sample size L for the evaluation of the level of significance.

## 2.3 Akaike Information Criterion Ratio test

Information criteria are useful tools for model selection. It is reasonable to retain that Akaike Information Criterion (*AIC*; Akaike, 1974) is the most famous among them. Based on the Kullbach-Leibler discrepance measure, if $\boldsymbol{\theta}$ is the parameter set of a *k*-dimensional model ($k = dim(\boldsymbol{\theta})$), AIC is defined as:

$$AIC = -2\ell(\widehat{\boldsymbol{\theta}}) + 2k. \tag{4}$$

The model that best fits data has the lowest value of *AIC* between candidates. It is useful to observe that the term proportional 135 to the number of model parameters allows to account for the increased estimator variance due to a larger parametrization and embodies the principle of parsimony.

Sugiura (1978) observed that *AIC* can lead to misleading results for small samples; he proposed a new measure for AIC:

$$AIC_c = -2\ell(\hat{\theta}) + \frac{2k(k+1)}{L-k-1} \tag{5}$$

where a second-order bias correction is introduced. Burnham e Anderson (2004) suggested to use this version only when 140 $L/k_{max} < 40$, being $k_{max}$ the maximum number of parameters between the compared models. However, for larger L, $AIC_c$ converges to AIC. For a quantitative comparison between *AIC* and $AIC_c$ in the extreme value stationary model selection framework see also Laio et al. (2009).

In order to select between stationary and nonstationary candidate models, we use the ratio

$$AIC_R = \frac{AIC_{ns}}{AIC_{st}}. \tag{6}$$





where the subscripts indicate the *AIC* value obtained for a stationary (*st*) and a non-stationary (*ns*) model, both fitted with maximum likelihood to the same data series.

Considering that the better fitting model has lower *AIC*, if the time series is non-stationary, the $AIC_R$ should be less than 1. Vice versa if the time series is stationary.

In order to provide a rigorous comparison between the use of *MK*, *LR* and $AIC_R$, we evaluated the $AIC_{R,\alpha}$ threshold value

corresponding to significance level $\alpha$ by numerical experiments.

More in detail, we adopted the following procedure:

1.  N = 10000 samples are generated from a stationary GEV parent distribution, with known parameters;

2.  for each of these samples the $AIC_R$ is evaluated, by fitting the stationary and non-stationary GEV models described in section 2.4, thus providing its empirical distribution (see pdf in fig. 1);

3.  exploiting the empirical distribution of $AIC_R$ the threshold associated with a significance level of $\alpha = 0.05$ is numerically evaluated: this value, $AIC_{R,\alpha}$, represents the threshold for rejecting the null hypothesis of stationarity (which in these generations is true) in 5% of the synthetic samples.

This procedure was applied both for *AIC* and $AIC_c$. The experiment was repeated for a few selected sets of the GEV parameters,

including different trend values, and different sample lengths, as detailed in section 3.

**2.4 The GEV parent distribution**

The cumulative Generalized Extreme Value (GEV) distribution (Jenkinson, 1955) can be expressed as:

$$F(z, \boldsymbol{\theta_{st}}) = \begin{cases} exp\left\{-\left[1 + \varepsilon\left(\frac{z-\zeta}{\sigma}\right)\right]^{-1/\varepsilon}\right\} & \varepsilon \neq 0 \\ exp\left\{-exp\left[-\left(\frac{z-\zeta}{\sigma}\right)\right]\right\} & \varepsilon = 0 \end{cases} \qquad \sigma > 0 \qquad (7)$$

where $\zeta, \sigma, \varepsilon$ are respectively known as the position, scale and shape parameter, $\boldsymbol{\theta_{st}} = [\zeta, \sigma, \varepsilon]$, is a general and comprehensive

way to express the parameter set in the stationary case. The flexibility of GEV in contemplating Gumbel, Fréchet and Weibull distributions (for $\varepsilon = 0, \varepsilon > 0$ and $\varepsilon < 0$ respectively) makes it eligible for a more general discussion about non-stationarity implications.

Traditional extreme value distributions can be used into a nonstationary framework, modelling their parameters as function of time or other covariates (Coles, 2001) producing $\boldsymbol{\theta_{st}} \rightarrow \boldsymbol{\theta_{ns}} = [\zeta_t, \sigma_t, \varepsilon_t]$.

In this study, only a linear trend with time *t* in the position parameter $\zeta$ has been introduced, leading Eq. (7) to be expressed as:





$$F(z, \boldsymbol{\theta}_{ns}) = \begin{cases} exp\left\{-\left[1 + \varepsilon\left(\frac{z-\zeta_t}{\sigma}\right)\right]^{-1/\varepsilon}\right\} & \varepsilon \neq 0 \\ exp\left\{-exp\left[-\left(\frac{z-\zeta_t}{\sigma}\right)\right]\right\} & \varepsilon = 0 \end{cases} \qquad \sigma > 0 \qquad (8)$$

with

$$\zeta_t = \zeta_0 + \zeta_1 t, \qquad (9)$$

and $\boldsymbol{\theta}_{ns} = [\zeta_0, \zeta_1, \sigma, \varepsilon]$.

It is important to note that Eq. (8) is a more general way to define the GEV and has the property of degenerating into Eq. (7) for $\zeta_1 = 0$: in other words Eq. (7) represents a nested model of Eq. (8) which confirms the suitability of the Likelihood Ratio test for model selection.

The estimation of GEV parameters is often performed by means of the L-moments (Hosking, 1990), linear combinations of

PWM (Hosking et al., 1985). Given a time series of values **z**, sorting it in ascending order, sample PWM can be expressed using the following relationships:

$$\beta_0 = \frac{1}{L}\sum_{j=1}^{L} z_j$$

$$\beta_1 = \frac{1}{L}\sum_{j=1}^{L} \frac{(j-1)}{(L-1)} z_j$$

Between PWM and L-moments the following relationships hold:


$$\lambda_1 = \beta_0 = \frac{1}{L}\sum_{j=1}^{L} z_j$$

$$\lambda_2 = 2\beta_1 - \beta_0 = 2\frac{1}{L}\sum_{j=1}^{L} \frac{(j-1)}{(L-1)} x_j - \frac{1}{L}\sum_{j=1}^{L} z_j$$

We observe that, imposing trend in the mean has reflections only in $\lambda_1$, and does not affect $\lambda_2$. The analytical proof based on sample relationships is provided in Appendix.

In this work we used the maximum likelihood method (ML) to estimate GEV parameters from sample data. ML allows to treat

$\zeta_1$ as an independent parameter, as well as $\zeta_0$, $\sigma$ and $\varepsilon$. To this purpose we exploited the R package extRemes (Gilleland and Katz, 2016).





**2.5 Numerical evaluation of test power and significance level**

The power of a test is related to the second type error and is the probability of correctly rejecting the null hypothesis when it

is false. In particular, defining α (level of significance) the probability of a Type I error and β the probability of a Type II error, we have $power = 1 - \beta$. The maximum value of power is 1, which correspond to $\beta = 0$, i.e. no probability of Type II error. A fair comparison between the null and the alternative hypotheses would see $\alpha = \beta = 0.05$, which provides power = 0.95. In most applications conventional values are $\alpha = 0.05$ and $\beta = 0.2$, meaning that a 1-to-4 trade-off between α and β is accepted. Thus, assuming a significance level 0.05, a power level less than 0.8 should be considered too low. For each of the tests described in

subsections 2.1, 2.2 and 2.3, the power was numerically evaluated according to the following procedure:

1) N = 2000 Monte Carlo synthetic series are generated using the non-stationary GEV in Eqs. (8-9) as parent distribution with fixed parameter set $\boldsymbol{\theta}_{ns} = (\zeta_0, \zeta_1, \sigma, \varepsilon)$ and length L, being $\zeta_1 \neq 0$.

2) The threshold $AIC_{R,\alpha}$ associated with a significance level $\alpha = 0.05$ is numerically evaluated, as described in section 2.3 using the corresponding parameter set $\boldsymbol{\theta}_{st} = (\zeta_0, \sigma, \varepsilon)$ of GEV parent distribution.

3) From these synthetic series the power of the test is estimated as:

$$rejection\ rate = \frac{N_{rej}}{N}$$

where $N_{rej}$ is the number of series for whom the null hypothesis is rejected, as in Yue et al. (2002a).

The same procedure, with N = 10000, was used in order to check the actual significance level of test, which is the probability

of first type error i.e. the probability of rejecting the null hypothesis when it is true. The task was performed by following the above steps from 1 to 3 while replacing $\boldsymbol{\theta}_{ns}$ with $\boldsymbol{\theta}_{st}$ at step 1), in such a case the rejection rate $N_{rej}/N$ represents the actual level of significance $\alpha$.

**3 Sensitivity analysis, results and discussion**

A comparative evaluation of the tests' performance was carried out for all the GEV parameter sets $\boldsymbol{\theta}_{ns}$ obtained considering

three values of ε (-0.4, 0, 0.4) and three values of σ (10, 15, 20). The position parameter was kept always constant and equal to $\zeta_0 = 40$. Then, for any possible couple of σ and ε, we considered $\zeta_1$ ranging from -1 to 1 with step 0.1. Such a range of parameters represents a wide domain in the hydrologically feasible parameters space of annual maximum daily rainfall. Upper bounded ($\varepsilon = -0.4$), EV1 ($\varepsilon = 0$), and heavy tailed ($\varepsilon = +0.4$) cases are included. Moreover, for each of these parameter sets $\boldsymbol{\theta}_{ns}$, N samples of different size (30, 50 and 70) were generated.

For a clear exposition of results, this section is divided into four subsections. In the first one we focus on the opportunity of using AIC or $AIC_c$ for the evaluation of $AIC_R$, in the second one the comparison of test power and its sensitivity analysis to parent distribution parameters and sample size is shown. In the third one, the evaluation of the level of significance for all tests





and in particular the validity of the $\chi^2$ approximation for the D statistic is discussed. In the fourth subsection the numerical investigation on the sample variability of parameters is reported.

### 3.1 Evaluation of $AIC_R$, with AIC or $AIC_c$

Considering the non-stationary GEV four-parameters model, in order to satisfy the relation $L/k_{max} < 40$ suggested by Burnham e Anderson (2004), a time series with a record length not less than 160 should be available. Following this simple reasoning the $AIC$ should be considered de facto not-applicable to any annual maximum series showing a changing point in the '70-80s (e.g. Kiely, 1999). In our numerical experiment, the second-order bias correction of Sugiura (1978) should be always used because for the maximum sample length, L = 70, we have $L/k_{max} = 70/4 = 17.5$ for the non-stationary GEV. Nevertheless, we checked if using $AIC$ or $AIC_c$ is important in such a use of the ratio $AIC_R$. To this purpose we evaluated from synthetic series the percentage differences between the power of $AIC_R$, evaluated by means of $AIC$ and $AIC_c$. In Fig. 2 the empirical probability density functions of such percentage differences, grouped according to sample length, are plotted for generations with ε = 0.4 and different values of σ. It is interesting to note that only for L = 70 the error distribution shows a regular and unbiased bell-shaped distribution. Then we observe for L = 50 a small negative bias (about -0.02%), while for L = 30 a bias of -0.08 with a multi-peak and negatively skewed pdf. The latter pdf also has a higher variance than the others. Similar results were obtained for all values of ε, providing a general amount of differences always very low and allowing to conclude that the use of $AIC$ or $AIC_c$ does not significantly affect the power of $AIC_R$ for the cases examined. This follows the combined effect of the sample size (whose minimum value considered here is 30) and the limited difference in the number of parameters in the selected models. In the following we will refer and show only the plots obtained for the $AIC_R$ in Eq. (6) with $AIC$ evaluated as in Eq. (4).

### 3.2 Dependence of power on parent distribution parameters and sample size

The effect of parent distribution parameters and sample size on the numerical evaluation of power and significance level of $MK$, $LR$ and $AIC_R$ for different values of $\varepsilon$, $\sigma$ and $\zeta_1$ is shown in Fig. 3. The curves represent both significance level which is shown for $\zeta_1 = 0$ (true parent is the stationary GEV) and power for all other values $\zeta_1 \neq 0$ (true parent is the non-stationary GEV). Each subplot in Fig. 3 shows the dependence on the trend coefficient of power and significance level of $MK$, $LR$ and $AIC_R$ for one set of parameter values and different sample sizes. In all subplots the test power strongly depends on trend coefficient and sample size. This dependence is also affected by parent parameter values. In all cases the power reaches 1 for strong trend and approaches 0.05 (the chosen level of significance) for weak trend ($\zeta_1$ close to 0). In all combinations of the shape and scale parameters, and expecially for short samples, for a wide range of trend values the power has values well below the conventional value 0.8. The curves' slope between 0.05 and 1 is sharp for long samples and slow for short samples. It also depends on the parameter set, being such a slope generally slower for higher values of the scale ($\sigma$) and shape ($\varepsilon$) parameters





of the parent distribution. Significant difference of power between $MK$, $LR$ and $AIC_R$ is observable when the sample size is smaller and still more when the parent is heavy tailed ($\varepsilon = +0.4$).

In particular, for $\varepsilon = 0$, -0.4 and $L = 50, 70$ it is possible to report a slightly larger power of $LR$ with respect to $AIC_R$ and $MK$, but values are very close to each other. Interesting is the reciprocal position of $MK$ and $AIC_R$ power curves: in fact, the $AIC_R$ power is always larger than the $MK$ one, except when $\varepsilon = -0.4$, without sensible influence of the scale parameter.

Higher difference is found for heavy tailed parent distribution ($\varepsilon = +0.4$). While $LR$ keeps having the largest power, the difference with respect to $AIC_R$ remains small while the $MK$'s power almost collapses to values always smaller than 0.5.

Practical consequences of such patterns are very important and are discussed in the conclusion section.

### 3.3 Sensitivity and evaluation of the actual significance level

We evaluated the threshold values (corresponding to a significance level of 0.05) for accepting/rejecting the null hypothesis of stationarity according to the methodologies recalled in subsections 2.1 and 2.2 for $MK$ and $LR$ tests and introduced in section 2.3 for $AIC_R$. Based on such thresholds we exploited the generation of stationary series ($\zeta_1 = 0$) in order to numerically

evaluate the rate of rejection of the null-hypothesis, i.e. the actual significance level of the tests considered in the numerical experiment, following the procedure described in subsection 2.5.

Table 1 shows the numerical values of the actual level of significance, obtained numerically, to be compared to the theoretical value 0.05 for all the considered sets of parameters and sample size. Among the three measures for trend detection the $LR$ shows the worst performance. Results in Table 1 show that the rejection rate of the (true) null hypothesis is systematically

higher than it should be, and it is also dependent on parent parameter values. Such effect is exalted when the parent distribution is upper bounded ($\varepsilon = -0.4$) and for higher values of the scale parameter. In practice this implies that when using the $LR$ test, as described in subsection 2.2, one actually has a probability of rejecting the true null hypothesis of stationarity quite higher than he knows.

On the other hand, the performances of $MK$ with respect to the designed level of significance are less biased and independent

from the parameter set. Similar good performances are trivially obtained for the $AIC_R$, whose rejection threshold is numerically evaluated.

The plot in Fig. 4 is displayed in order to focus on the actual value of the level of significance and in particular on the $LR$ approximation $D \sim \chi_m^2$ as a function of the sample length $n$. The difference between theoretical and numerical values of the significance level is represented by the distance between the bottom value of the curve (obtained for $\zeta_1 = 0$, i.e. the stationary

GEV model) and the chosen level of significance 0.05 which is represented by the horizontal dotted line. In particular in Fig. 4 results for the parameter set ($\sigma = 15, \varepsilon = -0.4$) show that the actual rate of rejection is always higher than the theoretical one and changes significantly with the sample size, which means that the $\chi_m^2$ approximation leads to significantly underestimating the rejection threshold of the D statistic. Moreover, it seems that the entire curves of the $LR$ power (in red) are upward translated as a consequence of the significance level overestimation, meaning that the $LR$ test power is also





overestimated because of the approximation $D \sim \chi_m^2$. These results suggest, for the *LR* test, the use of a numerical procedure (as the one introduced for $AIC_R$ in subsection 2.3) for evaluating the D distribution and the rejection threshold.

Other considerations can be made on the use of $AIC_R$. As explained in subsection 2.3 we empirically evaluated by numerical generations the $AIC_{R,\alpha}$ threshold value with significance level 0.05 for each of the parameter sets and sample sizes considered. Similar results were obtained using the $AIC_c$ which are not shown for brevity. We found a significative dependence of $AIC_{R,\alpha}$

on the sample size. Fig. 5 shows curves of $AIC_{R,\alpha}$ obtained for each of the parameter sets vs sample size. It is also worth noting that all curves asymptotically trend to 1 as *L* increase. This property is due to the structure of *AIC* and peculiarity of the nested models used in this paper: while using a sample generated with weak non-stationarity (i.e. when $\zeta_1 \rightarrow 0$ in Eq. (9)) the maximum likelihood of model (7), $\ell(\widehat{\boldsymbol{\theta}}_{st})$, tends to $\ell(\widehat{\boldsymbol{\theta}}_{ns})$ of model (8) leaving only the bias correction in *AIC* to be discriminant for model selection. As a consequence, $AIC_{R,\alpha}$ will be always lower than 1, but, increasing sample size, also both

the likelihood terms $-2\ell(\widehat{\boldsymbol{\theta}}_{st})$ and $-2\ell(\widehat{\widehat{\boldsymbol{\theta}}_{ns}})$ in Eq. (4) will increase, pushing $AIC_R$ toward the limit 1. On the other hand, Fig. 5 shows that the threshold value $AIC_{R,\alpha}$ is significatively smaller than 1 up to *L* values well beyond the length usually available in this kind of analysis. Hence the numerical evaluation of the threshold has to be considered a required task in order to provide an assigned significance level to model selection. On the other hand, the simple adoption of the selection criteria $AIC_R < 1$ (i.e. $AIC_{R,\alpha} = 1$), would correspond to an unknown significance level dependent on the parent distribution and

sample size. In order to highlight this point, we evaluated the significance level $\alpha$ corresponding to $AIC_{R,\alpha} = 1$ following the procedure described in subsection 2.5 by generating N = 10000 synthetic series for any parameter set and sample length. Results, provided in Tab. 2, show that, in the explored GEV parameter domain, $\alpha$ ranges between 0.16 and 0.26 mainly depending on the sample length and the shape parameter of the parent distribution.

### 3.4 Sample variability of parent distribution parameters

Results shown above, with regard to performances of parametric and non-parametric tests, are in our opinion quite surprising and important. It is proven that the preference widely accorded to non-parametric tests being their statistics allegedly independent from the parent distribution is not well founded. On the other hand, the use of parametric procedures raises the problem of correctly estimating the parent distribution and, for the purpose of this paper, its parameters. Moreover, as being the trend coefficient $\zeta_1$ a parameter of the parent distribution in non-stationary condition, the proposed parametric approach

provides a maximum likelihood-based estimation of the same trend coefficient which is hereafter called ML-$\zeta_1$. For a comparison with non-parametric approaches we also evaluated the sample variability of the Sen's slope measure (δ) of the imposed linear trend. In order to provide insights into these issues, from the same sets of generations exploited above, we also analysed the sample variability of the maximum likelihood estimates ML-$\varepsilon$ ML-$\sigma$, for different parameter sets and sample length.

We evaluated sample variability s[.], as the standard deviation of the ML estimates of parameter values obtained from synthetic series. Results are shown in Figs. 6 and 9, for different parameter sets and sample size, vs true $\zeta_1$ values. In Fig 6, on the first





subplots row we show s[ML-$\zeta_1$] and on the second row the Sen's slope median s[δ]. The sample variability of the linear trend is in both cases strongly dependent on sample size and independent from the true $\zeta_1$ value in the range examined $[-1,1]$. It reaches high values for short samples and in such cases also its dependence on the scale and shape parent parameters is relevant.

The ML estimation of the trend coefficient is always more efficient than Sen's slope and this is observed in particular for heavy tailed distributions.

In Figs. 7 and 8 we show the empirical distributions of the Sen's slope δ and ML-$\zeta_1$ estimates obtained from samples of size $L = 30$, providing a visual information about the range of trend values that may result from a local evaluation. Similar results, characterized by smaller sample variability shown in Fig. 6, are obtained for $L = 50$ and $L = 70$ and are not shown for brevity.

Fig. 9 shows the sample variability of ML-$\varepsilon$ and ML-$\sigma$, which is still independent from the true $\zeta_1$ for values of $\varepsilon = 0$ and 0.4 while for upper bounded GEV distributions ($\varepsilon = -0.4$) it shows a significant increase for higher values of $\sigma$ and high trend coefficients ($|\zeta_1| > 0.5$).

In order to better analyse such patterns, for the scale and shape parent parameters we report also the distribution of their empirical ML estimates for different parameter sets vs the true $\zeta_1$ value used in generation. The sample distribution of ML-$\varepsilon$

is shown in Fig. 10 for $L = 30$ and Fig. 11 for $L = 70$. The sample distribution of ML-$\sigma$ is shown in Fig. 12 for $L = 30$ and Fig. 13 for $L = 70$. Subplots show that the presence of a strong trend coefficient may produce significant loss in the estimator efficiency probably due to deviation from normal distribution of the sample estimates also for long samples. This suggests the need of more robust estimation procedures which provides higher efficiency for estimates of $\epsilon$ and $\sigma$ in case of strong observed trend.

## 335    4 Conclusions

The results shown have important practical implications. The dependence of power on the parent distribution parameters may significantly affect results of both parametric and non-parametric tests including the widely used Mann-Kendall.

For all the generation sets and tests conducted, under the null hypothesis of stationarity, the power has values ranging between the chosen significance level (0.05) and 1 for large (and larger) ranges of the trend coefficient. The test power always collapses

to very low values for weak (but climatically important) trend values (in the case of annual maximum daily rainfall, $\zeta_1$ equal to 0.2 or 0.3 mm per year, for example). In presence of trend, the power is also affected by the scale and shape parameters of the GEV parent distribution. This observation can be made with reference to samples of all the lengths considered in this paper (from 30 to 70 years of observation) but the use of smaller samples significantly reduces the test power and dramatically extends the range of $\zeta_1$ values for which the power is below the conventional value 0.8. The use of this sample size is not rare

considering that significative trends due to anthropic effects are typically investigated in periods following a changing point often observed in the '80s.

These results also imply that in spatial fields where the alternative hypothesis of non-stationarity is true, but the parent's parameters (including the trend coefficient) and the sample length are variable in space, the rate of rejection of the false null-



hypothesis may be highly variable from site to site and practically out of control. In other words, in such a case, the probability

of recognizing the alternative hypothesis of non-stationarity as true from a single observed sample may unknowingly change (between 0.05 and 1) from place to place. For small samples (as $L = 30$ in our analysis) and heavy tailed distributions, the power is always very low for all the investigated range of the trend coefficient.

Hence, considering the high spatial variability of the parent distribution parameters and the relatively short period of reliable and continuous historical observations usually available, a regional assessment of trend non-stationarity may suffer from the

different probability of rejection of the null hypothesis of stationarity (when it is false).

These problems affect, in slightly different measures, both parametric and non-parametric tests. While these considerations are generally applicable to all the tests considered, differences also emerge between them. For heavy tailed parent distributions and smaller samples, the *MK* test power decreases more rapidly than for the other tests considered. Low values of power are already observable for $L = 50$. The *LR* test slightly outperforms the $AIC_R$ for small sample size and higher absolute values of

the shape parameter. Nevertheless, the higher value of the *LR* power seems to be overestimated as a consequence of the $\chi^2_m$ approximation for the D statistic distribution (see section 3.3).

Results also suggest that theoretical distribution of the *LR* test-statistic based on the null hypothesis of stationarity may lead to a significative increase of the rejection rate compared to the chosen level of significance i.e. an abnormal rate of rejection of the null hypothesis when it is true. In this case the use of numerical techniques, based on the use of synthetic generations

performed by exploiting a known parent distribution, should be preferred.

By the light of these results we conclude that in trend detection on annual maximum series the assessment of the parent distribution and the choice of the null hypothesis should be considered fundamental preliminary tasks. According to this remark, it is advisable to make use of parametric tests by numerically evaluating both the rejection threshold for the assigned significance level and the power corresponding to alternative hypotheses. This also requires developing robust techniques for

individuation of the parent distribution and estimation of its parameters. To this perspective, the use of a parametric measure such as the $AIC_R$, may take account of different choices for the parent distribution and, even more important, allows to set a null hypothesis different from the stationary case, based on a priori information.

The need of robust procedures for assessing the parent distribution and its parameters is also proven by the numerical simulations we conducted. Sample variability of parameters (including the trend coefficient) may increase rapidly for series

with L as low as 30 years of annual maxima. Moreover, we observed that, in case of highest trend, numerical instability and non-convergence of algorithms may affect the estimation procedure for upper bounded and heavy tailed distributions. Nevertheless, the sample variability of the ML trend estimator was found always smaller than the Sen's slope sample variability. Finally, it is worth noting that also the non-parametric Sen's slope method, applied to synthetic series, showed dependence on the parent distribution parameters with sample variability higher for heavy tailed distributions.

This analysis shed lights onto important eventual flaws in the at-site analysis of climate change provided by non-parametric approaches. Both test power and trend evaluation are affected by the parent distribution as well as they are in parametric methods. It is not a case, in our opinion, that many technical studies conducted in years around the world, provide





inhomogeneous maps of positive/negative trends and large areas of stationarity characterized by weak trends that are considered not statistically significative. Analogous concerns about the use of statistical tests have been expressed by Serinaldi

et al. (2018).

As already said, an advantage of using parametric tests and numerical evaluation of the test-statistic distribution is given by the possibility of assuming a null hypothesis based on a preliminary assessment of the parent distribution including trend detection by evaluation of non-stationary parameters. This could lead to a regionally homogeneous and controlled assessment of both significance level and power in a fair mutual relationship. With respect to the estimation of parameters of the parent

distribution, results suggest that at site analysis may provide highly biased results. More robust procedures are necessary like hierarchic estimation procedures (Fiorentino et al., 1987) providing estimates of $\varepsilon$ and $\sigma$ from detrended series (Strupczewski et al., 2016; Kochanek et al., 2013). Considering the high spatial variability of sample length, trend coefficient, scale and shape parameters we believe that the application of well-known and developed regional methods for selection and assessment of the parent distribution could be easily and profitably exploited in the context of non-stationarity and climate change detection in

annual maximum series and will be tackled in future research.

**Appendix**

Let us consider two different but consequent years, t and t + 1, setting $z(t) = z_0 + \alpha t$, for $\lambda_1$ there is:

$$\lambda_1(t) = \frac{1}{L}\sum_{j=1}^{L} z_0 + \alpha t = \frac{1}{L}\sum_{j=1}^{L} z_0 + \frac{1}{L}\sum_{j=1}^{L} \alpha t$$

$$\lambda_1(t+1) = \frac{1}{L}\sum_{j=1}^{L} z_0 + \alpha(t+1) = \frac{1}{L}\sum_{j=1}^{L} z_0 + \frac{1}{L}\sum_{j=1}^{L} \alpha(t+1) =$$


$$= \frac{1}{L}\sum_{j=1}^{L} z_0 + \frac{1}{L}\sum_{j=1}^{L} \alpha t + \frac{1}{L}\sum_{j=1}^{L} \alpha$$

Subtracting side by side:

$$\lambda_1(t+1) - \lambda_1(t) = \frac{1}{L}\sum_{j=1}^{L} z_0 + \frac{1}{L}\sum_{j=1}^{L} \alpha t + \frac{1}{L}\sum_{j=1}^{L} \alpha - \frac{1}{L}\sum_{j=1}^{L} z_0 - \frac{1}{L}\sum_{j=1}^{L} \alpha t = \frac{1}{L}\sum_{j=1}^{L} \alpha$$

Which proves that a trend in the mean value produces a related trend in $\lambda_1$.

By using the same approach for $\lambda_2$, we observe that:


$$\lambda_2(t) = 2\sum_{j=1}^{L-1} \frac{(L-j)(z_j + \alpha t)}{L(L-1)} - \frac{1}{L}\sum_{j=1}^{L} (z_j + \alpha t)$$



$$\lambda_2(t+1) = 2\sum_{j=1}^{L-1}\frac{(L-j)[z_j + \alpha(t+1)]}{L(L-1)} - \frac{1}{L}\sum_{j=1}^{L}[z_j + \alpha(t+1)]$$

Subtracting side by side:

$$\lambda_2(t+1) - \lambda_2(t) = 2\sum_{j=1}^{L-1}\frac{(L-j)(z_j + \alpha t)}{L(L-1)} - 2\sum_{j=1}^{L-1}\frac{(L-j)[z_j + \alpha(t+1)]}{L(L-1)} - \frac{1}{L}\sum_{j=1}^{L}(z_j + \alpha t) + \frac{1}{L}\sum_{j=1}^{L}[z_j + \alpha(t+1)] =$$

$$= 2\sum_{j=1}^{L-1}\frac{(L-j)(z_j + \alpha t)}{L(L-1)} - 2\sum_{j=1}^{L-1}\frac{(L-j)(z_j + \alpha t + \alpha)}{L(L-1)} - \frac{1}{L}\sum_{j=1}^{L}(z_j + \alpha t) + \frac{1}{L}\sum_{j=1}^{L}(z_j + \alpha t + \alpha) =$$

$$= 2\sum_{j=1}^{L-1}\frac{(L-j)(z_j + \alpha t - \alpha - x_j - \alpha t)}{L(L-1)} + \frac{1}{L}\sum_{j=1}^{L}(z_j + \alpha t + \alpha - x_j - \alpha t) =$$

$$= -2\sum_{j=1}^{L-1}\frac{(L-j)\alpha}{L(L-1)} + \frac{1}{L}\sum_{j=1}^{L}\alpha = \frac{-2\alpha}{L(L-1)}\sum_{j=1}^{L-1}(L-j) + \frac{L\alpha}{L} = \frac{-2\alpha}{L(L-1)}\left[\sum_{j=1}^{L-1}L - \sum_{j=1}^{L-1}j\right] + \alpha =$$

$$= \frac{-2\alpha}{L(L-1)}\left[L(L-1) - \frac{L(L-1)}{2}\right] + \alpha = \frac{-2\alpha}{L(L-1)}\frac{L(L-1)}{2} + \alpha = -\alpha + \alpha = 0$$

Which proves that a trend in the mean value does not affect $\lambda_2$.

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





**Table 1: Actual level of significance of tests for different sample size, scale and shape parent parameters**

| | L = 30 | | | | | | | | |
| | $\varepsilon = -0.4$ | | | $\varepsilon = 0$ | | | $\varepsilon = +0.4$ | | |
| | $\sigma = 10$ | $\sigma = 15$ | $\sigma = 20$ | $\sigma = 10$ | $\sigma = 15$ | $\sigma = 20$ | $\sigma = 10$ | $\sigma = 15$ | $\sigma = 20$ |
|---|---|---|---|---|---|---|---|---|---|
| MK | 0.048 | 0.047 | 0.047 | 0.047 | 0.050 | 0.050 | 0.046 | 0.049 | 0.048 |
| $AIC_R$ | 0.050 | 0.046 | 0.052 | 0.051 | 0.052 | 0.045 | 0.052 | 0.054 | 0.051 |
| LR | 0.104 | 0.103 | 0.115 | 0.061 | 0.064 | 0.060 | 0.084 | 0.081 | 0.083 |

| | L = 50 | | | | | | | | |
| | $\varepsilon = -0.4$ | | | $\varepsilon = 0$ | | | $\varepsilon = +0.4$ | | |
| | $\sigma = 10$ | $\sigma = 15$ | $\sigma = 20$ | $\sigma = 10$ | $\sigma = 15$ | $\sigma = 20$ | $\sigma = 10$ | $\sigma = 15$ | $\sigma = 20$ |
|---|---|---|---|---|---|---|---|---|---|
| MK | 0.050 | 0.047 | 0.046 | 0.044 | 0.047 | 0.050 | 0.049 | 0.044 | 0.048 |
| $AIC_R$ | 0.053 | 0.053 | 0.046 | 0.051 | 0.051 | 0.057 | 0.050 | 0.050 | 0.053 |
| LR | 0.079 | 0.078 | 0.074 | 0.060 | 0.063 | 0.063 | 0.070 | 0.069 | 0.070 |

| | L = 70 | | | | | | | | |
| | $\varepsilon = -0.4$ | | | $\varepsilon = 0$ | | | $\varepsilon = +0.4$ | | |
| | $\sigma = 10$ | $\sigma = 15$ | $\sigma = 20$ | $\sigma = 10$ | $\sigma = 15$ | $\sigma = 20$ | $\sigma = 10$ | $\sigma = 15$ | $\sigma = 20$ |
|---|---|---|---|---|---|---|---|---|---|
| MK | 0.050 | 0.052 | 0.054 | 0.052 | 0.051 | 0.047 | 0.049 | 0.048 | 0.046 |
| $AIC_R$ | 0.047 | 0.051 | 0.051 | 0.058 | 0.058 | 0.052 | 0.050 | 0.054 | 0.051 |
| LR | 0.069 | 0.069 | 0.073 | 0.063 | 0.065 | 0.058 | 0.062 | 0.062 | 0.063 |


**Table 2: Actual level of significance of $AIC_R$ test for $AIC_{R,\alpha} = 1$**

| | $\varepsilon = -0.4$ | | | $\varepsilon = 0$ | | | $\varepsilon = +0.4$ | | |
| | $\sigma = 10$ | $\sigma = 15$ | $\sigma = 20$ | $\sigma = 10$ | $\sigma = 15$ | $\sigma = 20$ | $\sigma = 10$ | $\sigma = 15$ | $\sigma = 20$ |
|---|---|---|---|---|---|---|---|---|---|
| L = 30 | 0.246 | 0.254 | 0.261 | 0.188 | 0.191 | 0.181 | 0.220 | 0.221 | 0.215 |
| L = 50 | 0.213 | 0.209 | 0.206 | 0.171 | 0.175 | 0.170 | 0.188 | 0.207 | 0.195 |
| L = 70 | 0.192 | 0.192 | 0.201 | 0.168 | 0.169 | 0.173 | 0.184 | 0.204 | 0.184 |




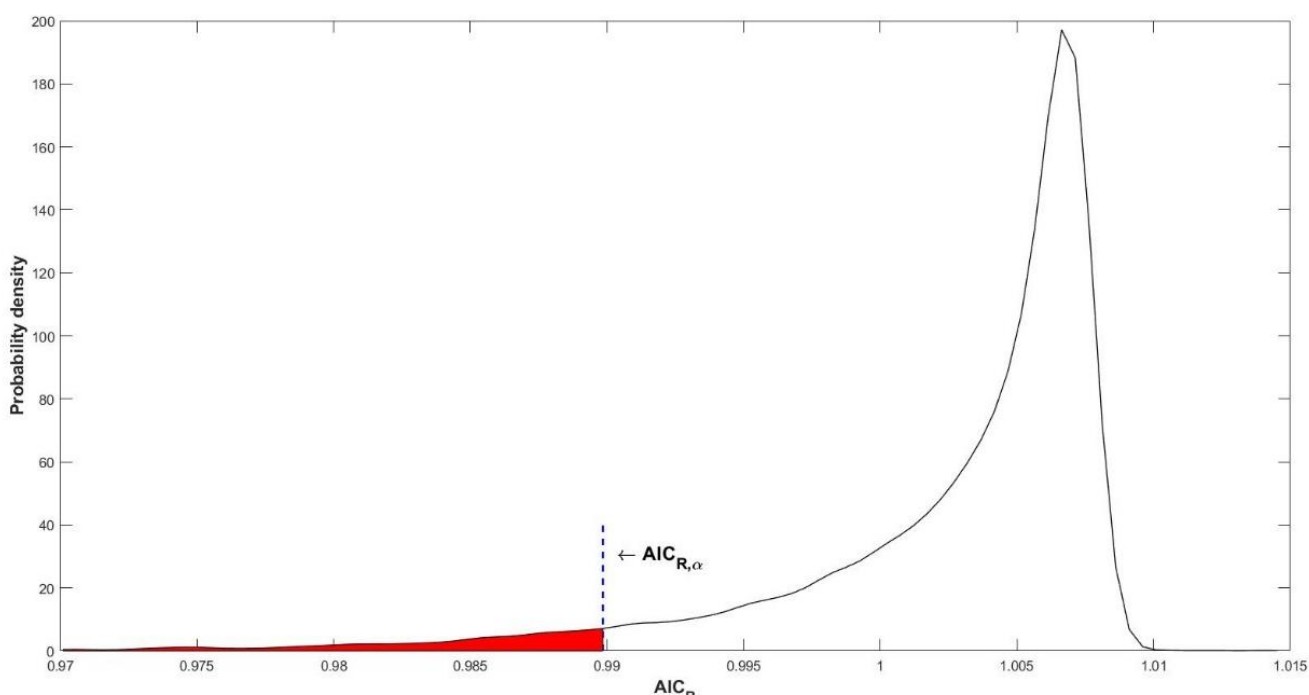

530        **Figure 1: Empirical distribution of AIC$_R$ and rejection threshold $AIC_{R,\alpha}$ of the null hypothesis (stationary GEV parent)**

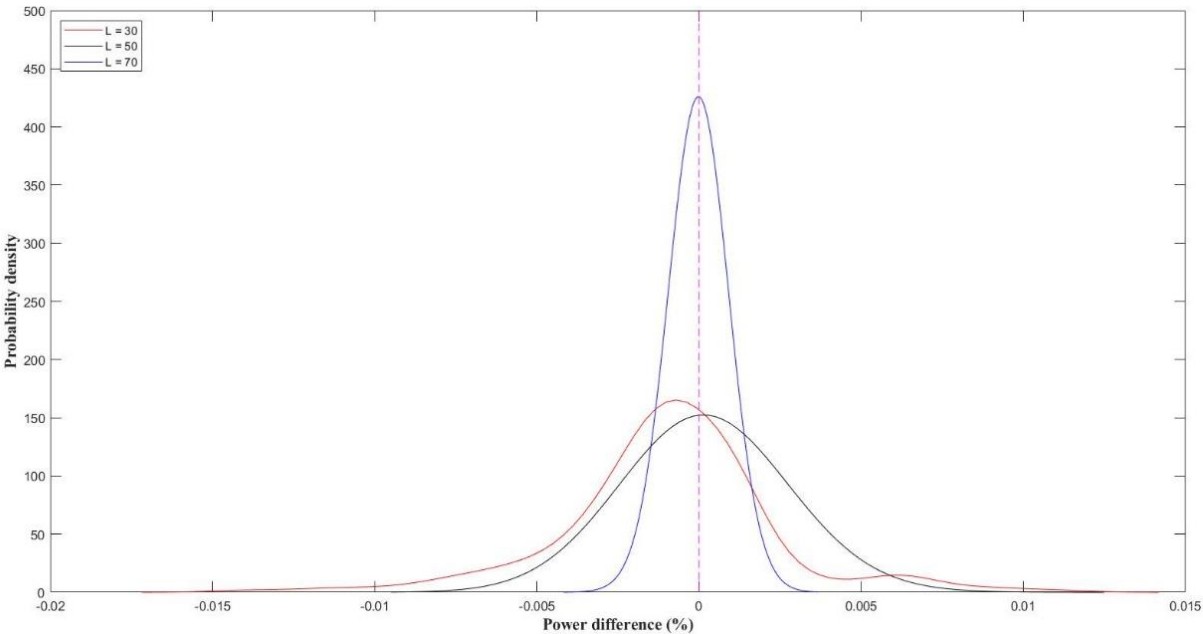

**Figure 2: Distributions of the differences between power of $AIC_R$ evaluated with $AIC$ and $AIC_c$ for $\varepsilon = 0.4$**





**Figure 3: Dependence of test power on trend coefficient, sample size, scale and shape parent parameters.**





**Figure 4: Focus on the actual level of significance reported for $\zeta_1 = 0$.**





**Figure 5:** $AIC_{R,\alpha}$ **thresholds for different parameter sets vs sample size**





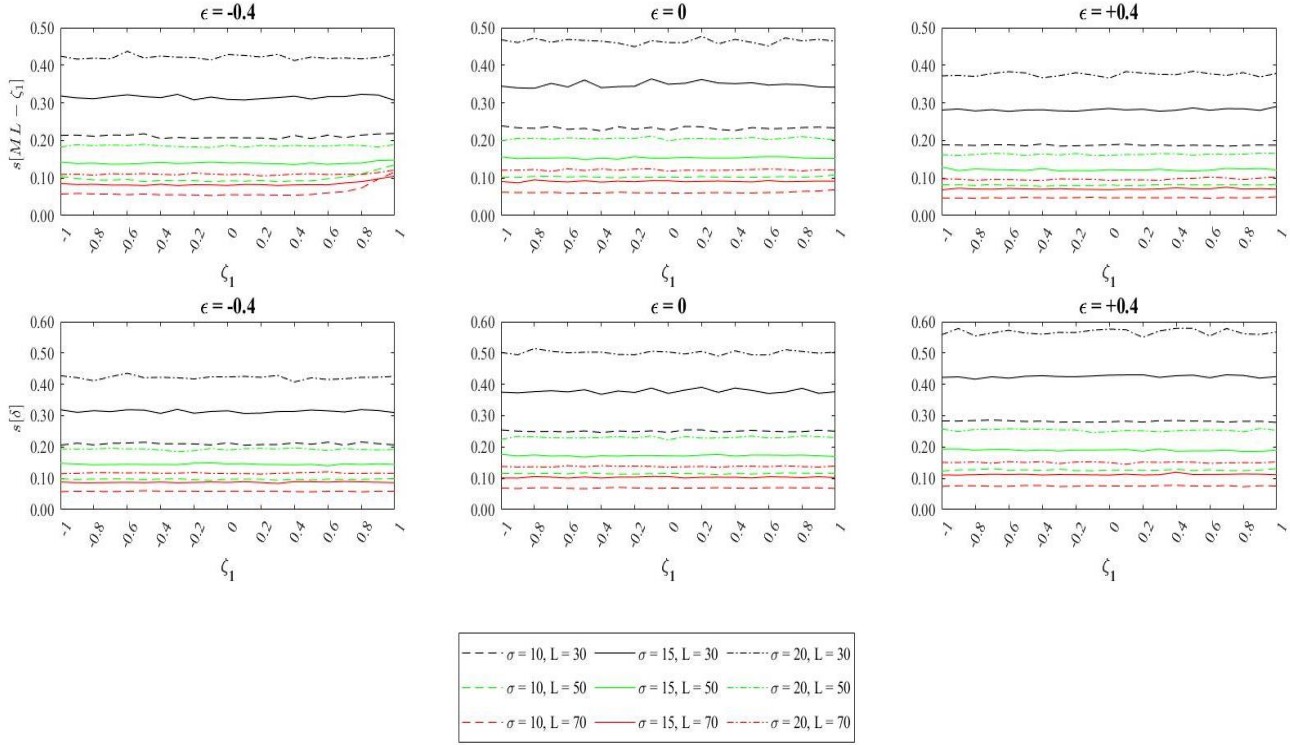

**Figure 6: Sample variability of ML-$\zeta_1$ and $\delta$ vs trend coefficient $\zeta_1$**



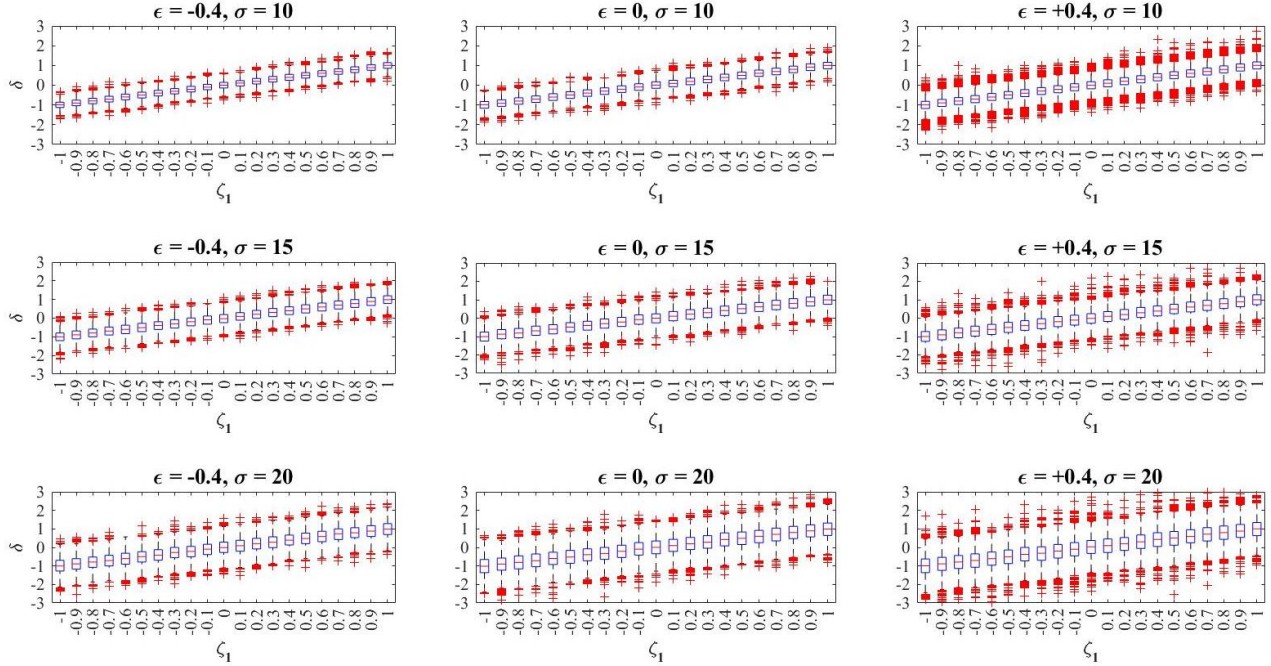


**Figure 7: Empirical distributions of $\delta$ evaluated from samples with $L = 30$, vs trend coefficient $\zeta_1$**

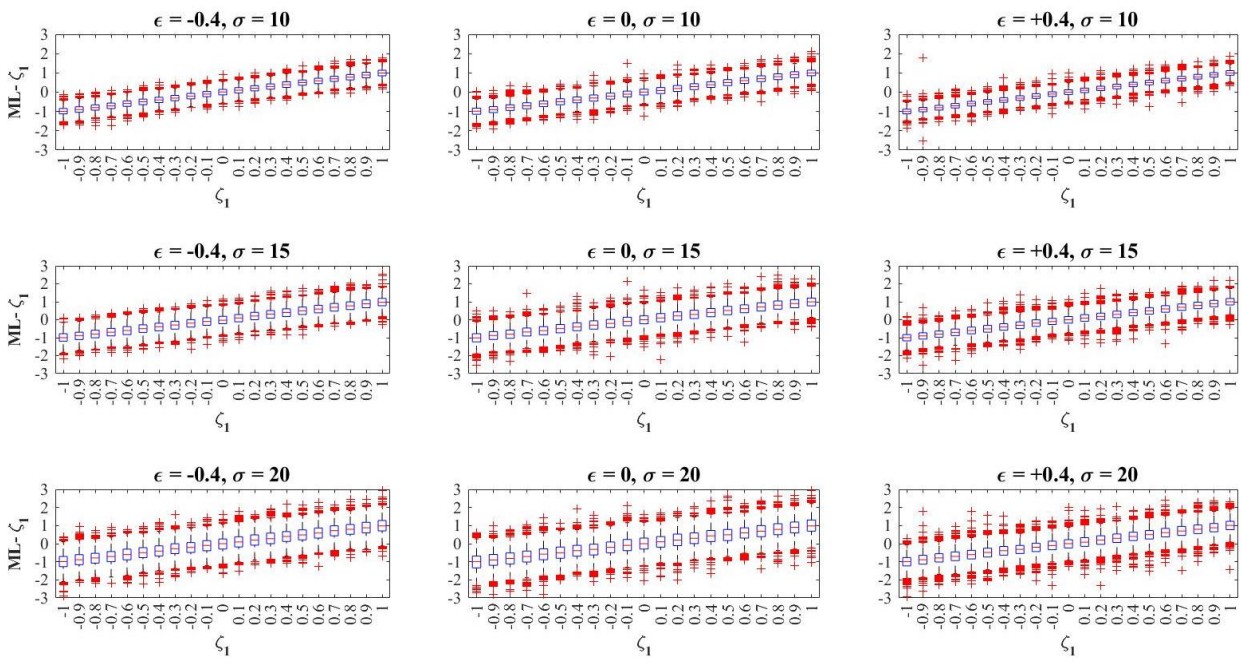

**Figure 8: Empirical distributions of ML-$\zeta_1$ evaluated from samples with $L = 30$ vs trend coefficient $\zeta_1$**





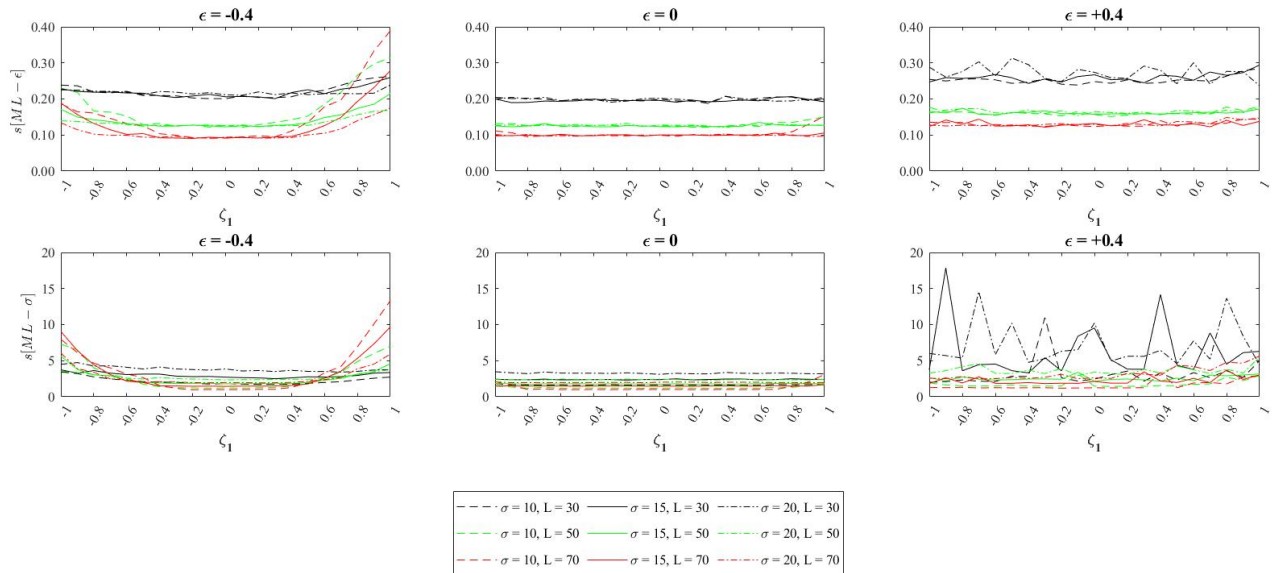


**Figure 9: Sample variability of ML-$\varepsilon$ and ML-$\sigma$ vs trend coefficient $\zeta_1$**

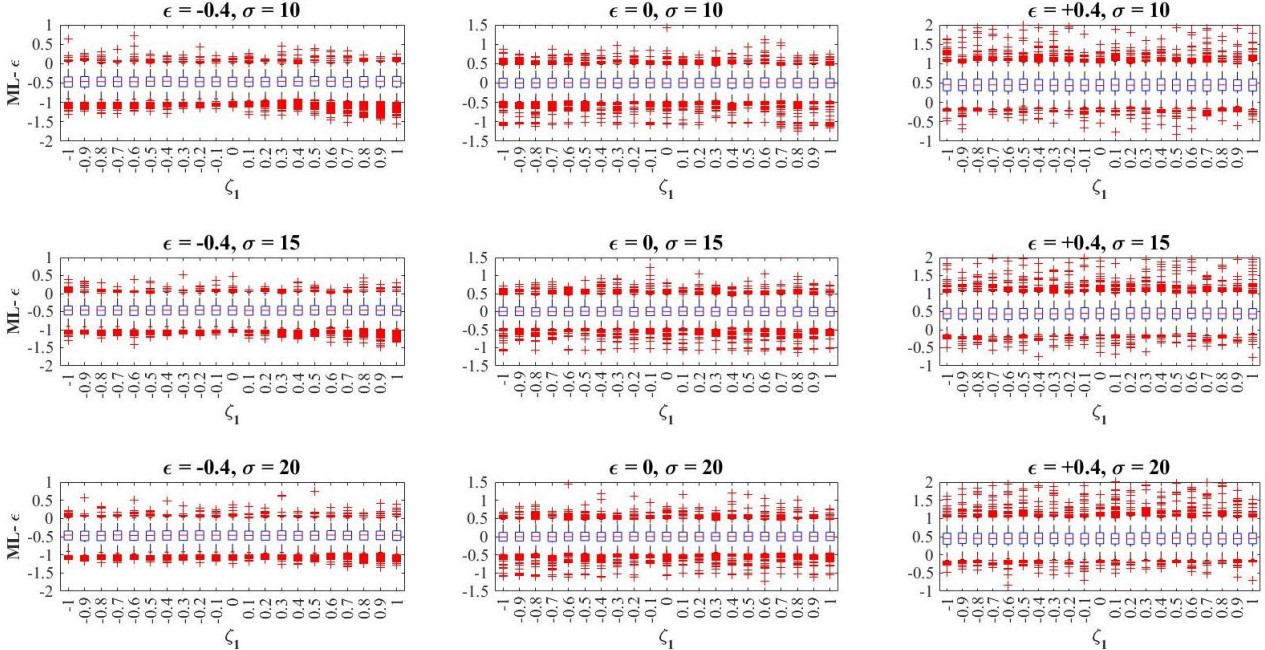

**Figure 10: Empirical distributions of ML-$\varepsilon$ evaluated from samples with $L = 30$ vs trend coefficient $\zeta_1$**






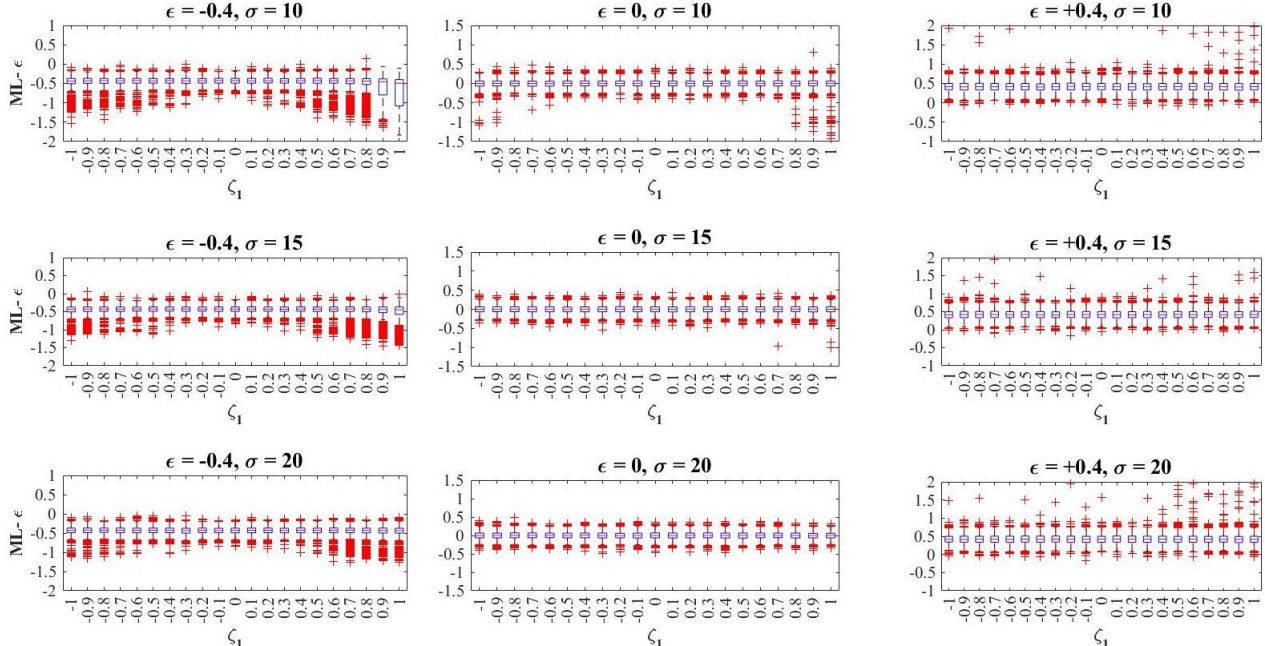

**Figure 11: Empirical distributions of ML-$\varepsilon$ evaluated from samples with $L = 70$ vs trend coefficient $\zeta_1$**

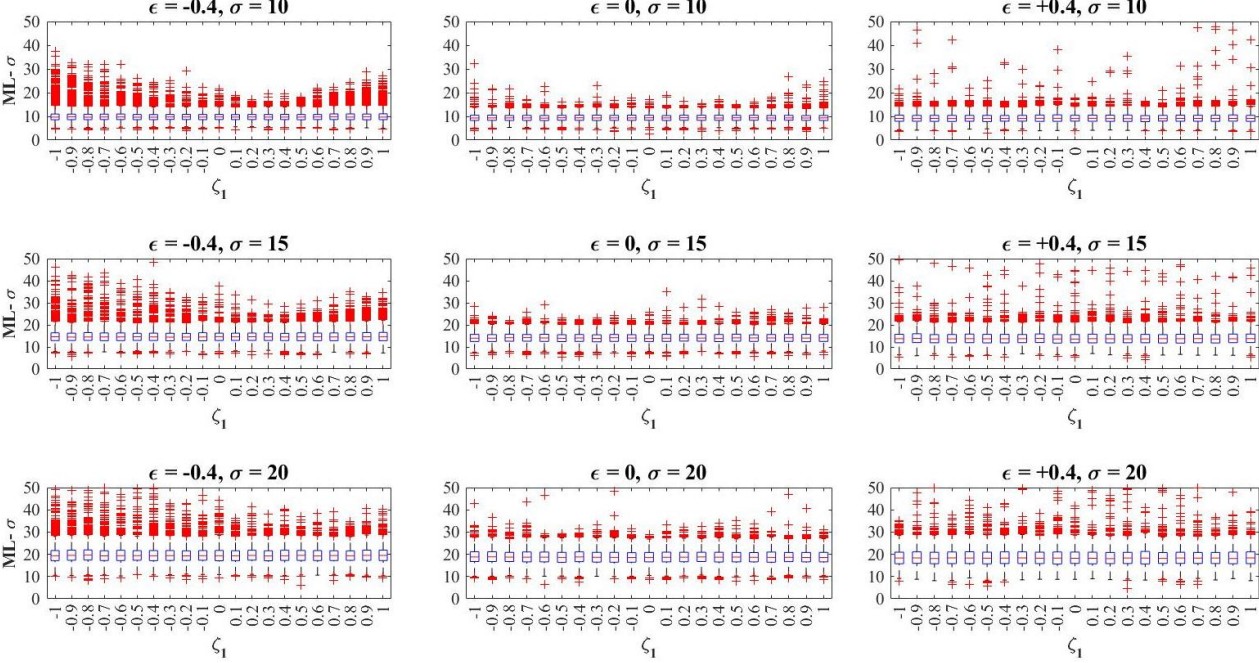

**Figure 12: Empirical distributions of ML-$\sigma$ evaluated from samples with $L = 30$ vs trend coefficient $\zeta_1$**






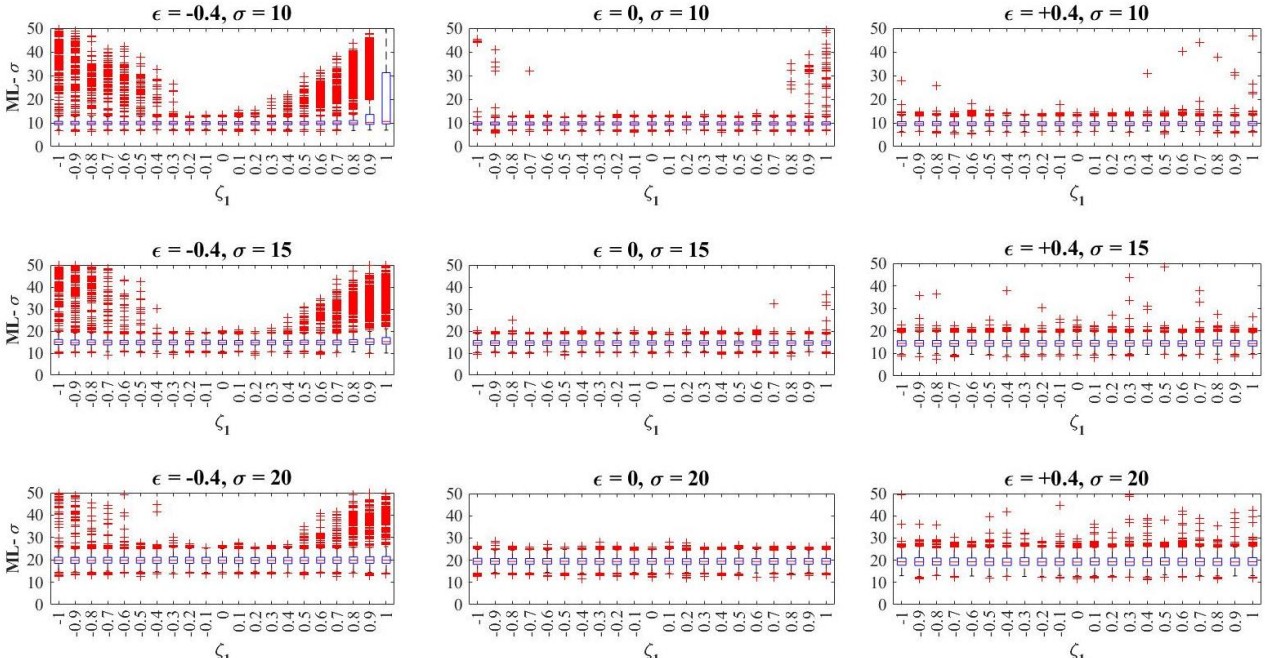

**Figure 13: Empirical distribution of ML-$\sigma$ evaluated from samples with $L = 70$**