# Peer review of "Numerical investigation on the power of parametric and nonparametric tests for trend detection in annual maximum series"

_Hydrology and Earth System Sciences, 2019_

## Referee Comment (RC1) · Anonymous Referee #1 · 9 Sep 2019

**General comment**

The manuscript investigates the power (and statistical significance) of non-parametric and parametric approaches for non-stationarity detection in annual maxima series. The methods analyzed in the work are the classical Mann-Kendall, likelihood ratio and AIC ratio tests; the investigation is performed thought a detailed numerical Monte Carlo experiment that makes use of the GEV distribution as a reference distribution. The experiment accounts for parent distribution parameter variability, including the parameter describing the non-stationary behavior of the average of the process, and sample variability. The topic is of paramount importance for hydrological applications and the conclusions drawn, which are clearly supported by the performed numerical experiments, give practical indications to practitioners about the application of the methods in real world problem, as discussed extensively in the conclusion section. In general, the manuscript is well written and organized; it deserves to be published in HESS. I have only a couple of suggestions to the Authors, as listed in the following.

- The first is about the assumptions made in this work as their significance with respect to natural phenomena; e.g. the non-stationary model accounts only for the variability of mean in time (which should be explicitly shown for the sake of clarity by reporting the theoretical expressions of the three first order moments a functions of the parameters), yet in nature the non-stationary behavior could imply also a variability in terms of the second order moment. Further, natural time series often depict dependence in time, which significantly affects the power of statistical methods for non-stationarity detection, as also recognized by the Authors themselves. I generally suggest the Authors to improve the discussion on practical limitations of those tests and of the conditions analyzed in their work, yet this is only a personal suggestion to improve the completeness of the discussion.

- Second, I would like to see a deeper comparison with previous literature works on the same topic; e.g. the Authors mention in the conclusion section the paper from Serinaldi et al. (2018), without giving further details. I believe that the comparison with previous literature results could help strengthen the general discussion presented in the conclusion section.

- Finally, the Authors should spend some efforts to improve the readability of the figures, e.g. by making the lines thick, by increasing the character size etc.
* * *

---

## Referee Comment (RC2) · Anonymous Referee #2 · 28 Sep 2019

The reviewed manuscript presents a thorough investigation, based on Monte Carlo simulations, of the power of parametric and non-parametric tests in detecting non-stationarities using annual maxima series. The study concludes that both types of tests exhibit significant deficiencies, displaying dependence of their power on the shape of the parent distribution, the sample length and trend intensity. Overall, I find the manuscript technically sound, logically organized and in general terms well written, deserving publication in HESS. Specific remarks on English usage and grammar are attached in the annotated .pdf of the manuscript, but further improvements are deemed necessary. The Authors are advised to seek help from a native English speaker with expertise in technical writing and scientific communication, or use language services. Some General comments are presented below:

1) The title of the manuscript is not informative of its content. I suggest "Monte Carlo investigation of the power of parametric and non-parametric tests for trend detection in annual maxima series", or something similar.

2) Lines 29, 147 and other parts of the manuscript: The statement attributed to Salas (1993) (see line 29) is theoretically incorrect. Stationarity is an attribute of stochastic processes (i.e. models) not of time-series (i.e. their realizations). More precisely, a stochastic process is said to be stationary, if and only if:

$$X_t \stackrel{\mathrm{d}}{=} X_{t+\tau}, \ \forall \ t, \tau$$

where $\stackrel{\mathrm{d}}{=}$ denotes equality in all finite-dimensional CDF's $F_{X,n}$, $n = 1, 2,....$ For example, $F_{X,1}(x; t) = F_{X,1}(x)$, $F_{X,2}(x_1, x_2; t_1, t_2) = F_{X,2}(x_1, x_2; |t_2\text{-}t_1|)$, $F_{X,3}(x_1, x_2, x_3; t_1, t_2, t_3) = F_{X,3}(x_1, x_2, x_3; |t_2\text{-}t_1|,|t_3\text{-}t_1|)$ and so on.

In the above context: a) lack of trends shifts and periodicities in a timeseries does not necessarily mean that the parent process is stationary, b) the wording in the manuscript should be properly modified to avoid use of the terms: "stationary timeseries" and "non-stationary timeseries"

3) The results presented in Figures 7-13 need to be discussed in more detail.

As a final remark, I think that in the concluding section, the Authors should at least comment on an important aspect related to the presented analysis: When inferring the properties of a stochastic process from data, one needs to analyze the available time-series assuming ergodicity. Since a non-stationary process is (by definition) non-ergodic, the stationarity assumption is central to any type of time-series analysis. Hence, non-stationary modeling of physical processes based on data (i.e. a single realization of a stochastic process) is theoretically inconsistent. That said, I believe that the findings of the Authors regarding uncertainty aspects of parametric and non-parametric tests in detecting non-stationarities, significantly underestimate those emerging when real world data is used.

[revised manuscript text omitted]

---

## Referee Comment (RC3) · Anonymous Referee #3 · 28 Sep 2019

The manuscript investigates the performances of parametric and non-parametric tests for trend detection. Specifically, the non-parametric Mann-Kendall test is compared with parametric Likelihood Ratio and Akaike Information Criterion Ratio tests. Analyses are conducted by Monte Carlo controlled experiments, using samples drawn from a GEV distribution with reliable parameters' values in representing daily-rainfall yearly maxima in Mediterranean climate.

Synthetic non-stationary time series were generated imposing a linear model (as a function of time) for the local GEV parameter, but keeping constant the shape and scale parameters, while for stationary time series all parameters were kept constant.

The analyses focus mainly in the evaluation of the power of the tests under the null hypothesis of stationarity. Results clearly quantify the degree of deficiency of the tests

in rejecting the null hypothesis in non-stationary samples, depending on various features, including the slope of the trend and the shape of the distribution, demonstrating also that performances of non-parametric tests can be affected by the shape of the underlying distribution of the sample.

**General comments**

The paper is timely and technically sound, and certainly of interest for HESS readers. Besides the specific comments listed below (which I consider minor by a technical point of view), the paper needs a careful proofreading for English language.

**Specific comments**

1) Lines 113-116. Description of Sen slope estimation and equation (2) should be revised: If N is the number of univocal (non-repeated) couples and $j$ is an index for the $j$-th couple $(x_i, x_k)$, why should be $j > k$? Maybe the authors mean $i > k$? Please check and better specify the role of $j$ index. Remove also "Sorting in ascending order ....", declaring that the median values is the final estimate is enough and the reader understand.

2) Lines 179-188 + Appendix. These lines + Appendix should be removed. All the analyses in the manuscript are based on ML estimates, thus there is no reason to keep a description of PWM and L-moments.

3) Section 2.5. It should be written that stationarity is assumed as null hypothesis (e.g. in line 207 and 210)

4) Section 2.5, lines 201 and 209. Just a curiosity: why experiments are conducted with a different number of samples (2000 vs 10000)?

5) Section 2.5, line 198. Authors can provide a reference for the sentence "1-to-4 trade-off between $\alpha$ and $\beta$ is accepted"?

6) Lines 214-219. I understand the choice of GEV parameters and it is reasonable to

my knowledge of rainfall maxima in Mediterranean climate. I was wandering whether it can be more informative to present results and figures in a more general way, e.g. as a function of the relative trend $\zeta_1/\zeta_0$ (which has dimension 1/time)?

7) Line 236. "a multi-peak . . ..": are you sure that it is not a sampling effect? Increasing the number of simulations is the result the same?

8) Lines 267-273 and Table 1. Similar to previous comment: are you sure that variability observed for different $\sigma$ (but keeping constant the other constraints) are not a sampling effect? Increasing the number of simulations is the result the same?

9) Line 301. I would specify here that series are stationary.

10) Section 3.4 (and maybe other parts of the manuscript, figures and tables). GEV parameters are always estimated by ML, thus I suggest to avoid the use of the prefix "ML-" before the symbol of the parameter (e.g. in Line 310 and 313). This would made more clear text, figure and tables.

11) Line 316. "Figs. 6 and 9": usually figures should be ordered as they are cited.

12) Figures. Please consider the opportunity to use larger fonts for labels, they are not readable here, and most probably Figure will be reduced in the final formatting.

13) Figure 6. Use the same range of scales (e.g. 0-0.6) in the y-axis for a fair comparison.

14) Figure 9. Again: are you sure that fluctuations are not due to sampling effects? See e.g. the subplot in the right part of the Figure 9.

15) Figures 7, 8, 10, 11, 12, 13 are not much informative. Please show only a selection of the most representative case. An additional option is to move these figures as supplementary material.

363, 2019.

---

## Author Comment (AC1) · 29 Oct 2019

**Authors' response to comments from Referee #1**

The authors gratefully acknowledge the anonymous referee for his positive review and remarkable insight proposals. In what follows in *Italic* are the comments provided by the Referee, and in **bold** fonts the authors' response. Changes to the manuscript are quoted and reported in ***"bold italic"*** font. Please consider that text here reported is still undergoing for the check from a professional native English speaker.

*The first is about the assumptions made in this work as their significance with respect to natural phenomena; e.g. the non-stationary model accounts only for the variability of mean in time (which should be explicitly shown for the sake of clarity by reporting the theoretical expressions of the three first order moments a functions of the parameters), yet in nature the non-stationary behavior could imply also a variability in terms of the second order moment. Further, natural time series often depict dependence in time, which significantly affects the power of statistical methods for non-stationarity detection, as also recognized by the Authors themselves. I generally suggest the Authors to improve the discussion on practical limitations of those tests and of the conditions analyzed in their work, yet this is only a personal suggestion to improve the completeness of the discussion.*

**Authors wish to thank the reviewer for these useful suggestions. We introduced in section 2.4 the theoretical expressions by Muraleedharan et al. (2010) of the three first order moments as functions of the parameters. We enlarged the discussion regarding general limits of statistical methods for detection of non-stationarity in both the introduction and the conclusive sections. In particular we specify, also dealing with the final remark from Referee #2, that our purpose here is to show that, in some cases, even a weak linear trend in the mean suffices to reduce power to unacceptable values, then, we decided to limit the investigation to variability of mean in time. Nevertheless, when dealing with natural series, a number of potential other sources of uncertainty, including variability in time of the second order moment, should be considered as we have remarked in the conclusions of the revised manuscript.**

**As main changes to the manuscript, in the introduction the following lines were introduced:**

***"The use of null hypothesis significance tests for trend detection has raised concerns and severe criticisms in a wide range of scientific fields as outlined by Vogel et al. (2013). Serinaldi et al. (2018) also provided an extensive critical review focusing on logical flaws and misinterpretations often related to their misuse."***

**and later,**

***"Nevertheless, as claimed by different authors (Milly et al. 2015, Beven, 2016, among others) the importance of power in earth system sciences fields has been largely overlooked in years while a strong attention is always given to the level of significance (i.e. type I error). As pointed out by Vogel et al. (2013) "a type II error in the context of an infrastructure decision implies under-preparedness, which is often an error much more costly to society than the type I error (overpreparedness)".***

**For changes to conclusions please see response to final remark from Referee #2.**

*Second, I would like to see a deeper comparison with previous literature works on the same topic; e.g. the Authors mention in the conclusion section the paper from Serinaldi et al. (2018), without giving further details. I believe that the comparison with previous literature results could help strengthen the general discussion presented in the conclusion section*

**We accepted this suggestion, we decided to move the comparison with previous literature in the introduction. This includes a more specific reference to the work from Serinaldi et al (2018) and also a more general discussion about the use of statistical power for trend detection in hydrology (see response to point 5) of Referee #3 for changes to manuscript).**

**Moreover, in the conclusion section, we introduced some final remarks about the lack of ergodicity to be considered when dealing with nonstationary stochastic process (see response to final remark of Referee #2 for changes to manuscript).**

*Finally, the Authors should spend some efforts to improve the readability of the figures, e.g. by making the lines thick, by increasing the character size etc.*

**Suggestion accepted.**

---

## Author Comment (AC2) · 29 Oct 2019

**Authors' response to comments from Referee #2**

The authors would like to thank the second reviewer for the positive overall evaluation and the constructive general comments. Suggestions and specific remarks on English usage and grammar are greatly appreciated and will be thoroughly accounted for.

In what follows in *Italic* are the comments provided by the Referee, and in **bold** fonts the authors' response. Changes to the manuscript are quoted and reported in ***"bold italic"*** font. Please consider that text here reported is still undergoing for the check from a professional native English speaker.

*The title of the manuscript is not informative of its content. I suggest "Monte Carlo investigation of the power of parametric and non-parametric tests for trend detection in annual maxima series", or something similar.*

**We thank the reviewer for this comment and for the suggested title we slightly changed it "Numerical investigation on the power of parametric and non-parametric tests for trend detection in annual maxima series"**

*Lines 29, 147 and other parts of the manuscript: The statement attributed to Salas (1993) (see line 29) is theoretically incorrect. Stationarity is an attribute of stochastic processes (i.e. models) not of time-series (i.e. their realizations). More precisely, a stochastic process is said to be stationary, if and only if:*

$$X_t \overset{\mathrm{d}}{=} X_{t+\tau}, \ \forall \ t, \tau$$

*where $\overset{\mathrm{d}}{=}$ denotes equality in all finite-dimensional CDF's $F_{X,n}$, n = 1, 2,.... For example, $F_{X,1}(x; t)= F_{X,1}(x)$, $F_{X,2}(x_1, x_2; t_1, t_2) = F_{X,2}(x_1, x_2; |t_2-t_1|)$, $F_{X,3}(x_1, x_2, x_3; t_1, t_2, t_3) = F_{X,3}(x_1, x_2, x_3; |t_2-t_1|,|t_3-t_1|)$ and so on.*

*In the above context: a) lack of trends shifts and periodicities in a timeseries does not necessarily mean that the parent process is stationary, b) the wording in the manuscript should be properly modified to avoid use of the terms: "stationary timeseries" and "non-stationary timeseries"*

**We thank the reviewer for this important comment. The statement which is recognized as "theoretically incorrect" is a literal quote from Chapter 18 of Handbook of Hydrology (Maidment, 1993). As mentioned by Koutsoyiannis and Montanari (2015), in their paper section "Semantic and historical review" about the concept of stationarity, this is not the only case. Also the Kendall and Stuart's book make reference to "stationary series", for not counting the large number of papers that may arise from such a keyword search in a database such as Scopus. While we recognize the huge importance of semantic consistency in the scientific literature, we observe that in our paper there is little chance for misconception if considering that we are working in the framework of a Monte Carlo experiment by using time series generated from theoretical models. Nevertheless, in order to avoid any possible confusion, we accepted the reviewer's suggestion by removing the quoted sentence in the**

**introduction (line 29) and introducing a more formal definition of stationarity. Then, we have checked throughout the manuscript for the use of "stationary (and nonstationary) time series". We found, besides the line 147 that was indicated by the reviewer, only one other "suspicious case" at line 264. In lines 147 we rephrased "if the time series is non-stationary, […]. Vice versa if the time series is stationary" into "if the time series arises from a non-stationary process, […]. Vice versa if the process is stationary"; Line 264 "the generation of stationary series …" into "the generation of series from a stationary model…".**

*The results presented in Figures 7-13 need to be discussed in more detail.*

**We have introduced some more description of such results, nevertheless, following comment 15) from Referee #3, we have also reduced the number of figures and subplots, limiting their display to representative selected cases.**

**The following comment was added to the revised manuscript**

*Subplots show that the presence of a strong trend coefficient may produce significant loss in the estimator efficiency probably due to deviation from normal distribution of the sample estimates also for long samples. This suggests the need of more robust estimation procedures which provides higher efficiency for estimates of $\epsilon$ and $\sigma$ in case of strong observed trend. It should be highlighted that efficiency in parameter estimation increases with sample size for $\varepsilon=[0,0.4]$, while it decreases for both $\varepsilon$ and $\sigma$, in the case $\varepsilon=[-0.4]$, where the trend of the location parameter implies a shift in time of the distribution upper bound.*

**See response to comment 15) from Referee #3 regarding the reduction of figures and subplots.**

*As a final remark, I think that in the concluding section, the Authors should at least comment on an important aspect related to the presented analysis: When inferring the properties of a stochastic process from data, one needs to analyze the available time-series assuming ergodicity.*

*Since a non-stationary process is (by definition) non-ergodic, the stationarity assumption is central to any type of time-series analysis. Hence, non-stationary modeling of physical processes based on data (i.e. a single realization of a stochastic process) is theoretically inconsistent. That said, I believe that the findings of the Authors regarding uncertainty aspects of parametric and non-parametric tests in detecting non-stationarities, significantly underestimate those emerging when real world data is used.*

**Also this comment is particularly welcome because it provide us with the possibility to share our general perspectives on issues concerning real data analysis. Ergodicity, in fact, is not only an important theoretical property of stationary stochastic processes but it also affects practical inferential tasks. Then, we added a final remark about different sources of uncertainty and perspectives about data usage and exogenous information exploitation to be used in environmental change modeling.**

**The following lines were added in conclusions:**

**"*As a final remark, concerning real data analysis, in our numerical experiment we showed that, in some cases, even a weak linear trend in the mean suffices to reduce power to unacceptable values. Yet we explored the simplest nonstationary working hypothesis by introducing a*

*deterministic linear dependence on time of the location parameter of the parent distribution. Obviously, when making inference from real observed data other sources of uncertainty may affect statistical inference (trend, heteroscedasticity, persistence, nonlinearity, etc), and moreover, if considering a nonstationary process with underlying deterministic dynamics, the process turns out to be non-ergodic, implying that statistic inference from sampled series is not representative of the process's ensemble properties (Koutsoyiannis and Montanari, 2015).*

*As a consequence, while considering a nonstationary stochastic process as produced by a combination of a deterministic function and a stationary stochastic process, other sources of information and deductive arguments should be exploited in order to identify the physical mechanism underlying such relationships. Even in such a case observed time series have a crucial role in order to calibrate and validate deterministic modeling or, in other words, for confirming or disproving the model hypotheses.*

*In the field of frequency analysis of extreme hydrological events, considering the high spatial variability of sample length, trend coefficient, scale and shape parameters, etc, we believe that physically based probability distributions could be further developed and profitably exploited for selection and assessment of the parent distribution in the context of non-stationarity and change detection in annual maximum series. Physically based probability distributions we refer to are: (i) those arising from stochastic compound processes introduced by Todorovic and Zelenhasic (1970), which include also the GEV (see Madsen et al., 1997) and the TCEV (Rossi et al., 1984), and (ii) the theoretically derived distributions following Eagleson (1972) whose parameters are provided by clear physical meaning and are usually estimated with support of exogenous information in regional methods (e.g. Gioia et al., 2008; Iacobellis et al., 2011; see also for a more extensive overview Rosbjerg et al., 2013).*

*Hence, we believe that "learning from data" will remain in future years a fundamental task for hydrologists facing the challenge of consistently identifying both deterministic and stochastic components of change. This involves crucial and interdisciplinary research to be developed in order to exploit as much information as possible while finding consistent frameworks for enhancing data analysis and physical knowledge to reduce uncertainty of prediction in a changing environment."*

---

## Author Comment (AC3) · 29 Oct 2019

**Authors' response to comments from Referee #3**

The authors would like to thank also the third reviewer for the kind attention and the significative and constructive observations and suggestions.

As a first general remark he/she suggests a careful proofreading for English language which will be certainly done on the revised version of the manuscript from a professional native English speaker.

Specific comments will be addressed in the following lines according to the same numbering provided by the reviewer:

In what follows in *Italic* are the comments provided by the Referee, and in bold **fonts** the authors' response. Changes to the manuscript are quoted and reported in ***"bold italic"*** font. Please consider that text here reported is still undergoing for the check from a professional native English speaker.

*1) Lines 113-116. Description of Sen slope estimation and equation (2) should be revised: If N is the number of univocal (non-repeated) couples and j is an index for the j-th couple (x_i, x_k), why should be j > k? Maybe the authors mean i > k? Please check and better specify the role of j index. Remove also "Sorting in ascending order ...." , declaring that the median values is the final estimate is enough and the reader understand.*

**Suggestion accepted.**

*2) Lines 179-188 + Appendix. These lines + Appendix should be removed. All the analyses in the manuscript are based on ML estimates, thus there is no reason to keep a description of PWM and L-moments.*

**Suggestion accepted, the appendix and the expressions of L-moments were removed and replaced by theoretical expressions of moments as from first comment from Referee #1.**

3) *Section 2.5. It should be written that stationarity is assumed as null hypothesis (e.g. in line 207 and 210)*

**Suggestion accepted, we modified lines 205 and 211 by adding, respectively, that "the null hypothesis of stationarity is false" and "the null hypothesis of stationarity (which) is true".**

*4) Section 2.5, lines 201 and 209. Just a curiosity: why experiments are conducted with a different number of samples (2000 vs 10000)?*

**We used N = 10000 generations to be sure that the effect of sample variability is negligible when estimating the real level of significance and the corresponding threshold of acceptance. For the evaluation of power (i.e. in all generations with values of $\zeta_1 \neq 0$) we found N = 2000 is a good compromise between quality of results and reasonable computational time. To this purpose we performed sample checks with N = 10000 obtaining negligible differences. We did not report such details for the sake of simplicity and also because power values obtained with generations with $\zeta_1 \neq 0$ are only shown in graphical form and as such they are not even distinguishable from those shown. Moreover, N = 2000 is the same number of generations**

**used by Yue et al. (2002a). In the revised manuscript we added this short explanation about this point in section 2.5:**

*"We used a reduced number of generations (N = 2000) for the evaluation of power as good compromise between quality of results and computational time and, also in analogy with Yue et al. (2002a)."*

**Regarding the number of generations N, see also answers to points 7), 8) and 14).**

*5) Section 2.5, line 198. Authors can provide a reference for the sentence "1-to-4 trade-off between α and β is accepted"?*

**Such question is everything but trivial, we can provide a reference about this (Cohen, 1994) and we introduced it in the revised manuscript, nevertheless such paper does not come from hydrological literature but from Psychology and is by far the most cited paper in Scopus about "statistical power". Reflecting about this point has stimulated a discussion about the apparent lack of references of this type in the earth system sciences that we have added in the conclusion section. The following lines were added in the conclusion section:**

*"Considering the feasibility of numerical evaluation of power, allowed by the parametric approach, we observe that, while the awareness of the crucial role of type II error is growing in latest years in the hydrological literature, a common debate would deserve more development about which power values should be considered acceptable. Such an issue is much more enhanced in other scientific fields where the experimental design is traditionally required to estimate the appropriate sample size to adequately support results and conclusions. In psychological research, Cohen (1994) proposed 0.8 as a conventional value of power, to be used with level of significance 0.05 thus leading to a ratio 4 to 1 between the risk of type II and type 1 error. The conventional value proposed by Cohen (1994) has been taken as reference by thousands of papers in social and behavioural sciences. In pharmacological and medical research, depending on real implications and nature of the type II error, conventional values of power may be as high as 0.999. This is the value suggested by Liebher (1990), when testing a treatment for patients' blood pressure. He stated, while "guarding against cookbook application of statistical methods", that "it should also be noted that, at times, type II error may be more important to an investigator then type I error".*

*We believe that, selecting between stationarity and non-stationarity models for extreme hydrological event prediction, a fair comparison between the null and the alternative hypotheses as α = β = 0.05 should be taken, which provides power = 0.95. In our discussion we considered 0.8 as minimum threshold for acceptable power values."*

*6) Lines 214-219. I understand the choice of GEV parameters and it is reasonable to my knowledge of rainfall maxima in Mediterranean climate. I was wandering whether it can be more informative to present results and figures in a more general way, e.g. as a function of the relative trend $\zeta_1/\zeta_0$ (which has dimension 1/time)?*

**This comment is particularly important and stimulating. Nevertheless, such generalization is out of the purpose of this paper and would require more extensive investigation. In facts, presenting results in terms of the ratio $\zeta_1/\zeta_0$ would make sense if results of the analyses were the same for different couples of $\zeta_1$ and $\zeta_0$ values producing the same ratio. Actually, this is not the case when $\sigma$ is fixed. We believe that invariant properties of the frequency distribution of rainfall or floods annual maximum values could be exploited for a generalization of these**

analyses, but this would involve consideration of scaling features of different order moments. This could be a quite interesting future development of this study, involving also time dependence of the scale parameter. Not any change has been done to the manuscript.

*7) Line 236. "a multi-peak . . ..": are you sure that it is not a sampling effect? Increasing the number of simulations is the result the same?*

In this figure we show some distributions of sampled errors in terms of difference between the power obtained with AIC and the power obtained with $AIC_c$. The curves show that the entire range of sampled errors provides negligible values compared to the expected power values. The different peaks in one curve (L = 30) are observed because we merged sample errors obtained for different values of $\sigma$ characterized by a small different (random) bias. We added such considerations in the revised paper:

*"Aim of this figure is to show that the difference between the power obtained with AIC and the power obtained with AICc is negligible. Anyway, different peaks in one curve (L = 30) can be explained by merging sample errors obtained for different values of σ."*

*8) Lines 267-273 and Table 1. Similar to previous comment: are you sure that variability observed for different σ (but keeping constant the other constraints) are not a sampling effect? Increasing the number of simulations is the result the same?*

Results shown in Table 1 are, to some extent, affected by a sampling effect. We had numerically checked the sample variability of the actual level of significance, which is quite smaller than the difference between the designed (0.05) and the actual value of the significance level for the LR test. We didn't report such analysis for the sake of simplicity nevertheless we used a very high number of generations (N = 10000) to produce these values (see also response to comment #4). On the other hand, we agree that there is not such evidence with specific reference to different σ values, while ε and L mostly affect results and accordingly we revised the manuscript rephrasing the sentence as:

*"Such effect is exalted when the parent distribution is upper bounded (ε=-0.4) and for shorter series (L = 30)."*

*9) Line 301. I would specify here that series are stationary.*
**Suggestion accepted**

*10) Section 3.4 (and maybe other parts of the manuscript, figures and tables). GEV parameters are always estimated by ML, thus I suggest to avoid the use of the prefix "ML-" before the symbol of the parameter (e.g. in Line 310 and 313). This would made more clear text, figure and tables.*

We would rather maintain the use of the prefix "ML-" in order to distinguish between values (ML-epsilon, ML-sigma, ML-z1) estimated from the series and the theoretical values (EPS, sigma, z1) used in the parent distribution. Not any change was done.

*11) Line 316. "Figs. 6 and 9": usually figures should be ordered as they are cited.*

**Suggestion accepted by revising the text by removing such reference to fig. 9 which is introduced later.**

*12) Figures. Please consider the opportunity to use larger fonts for labels, they are not readable here, and most probably Figure will be reduced in the final formatting.*

**Suggestion accepted**

*13) Figure 6. Use the same range of scales (e.g. 0-0.6) in the y-axis for a fair comparison.*

**Suggestion accepted**

*14) Figure 9. Again: are you sure that fluctuations are not due to sampling effects? See e.g. the subplot in the right part of the Figure 9.*

**We checked such results with different sets of random generations also increasing N up to 10000. Qualitatively results do not change. "*Randomness of results for L = 30 and σ = 15, 20 is probably due to a reduced efficiency of the algorithm that maximizes the log-Likelihood function, for heavy tailed distributions.*" Such consideration was added in the revised manuscript (Fig. 9 is now Fig. 8, because of the following comment).**

*15) Figures 7, 8, 10, 11, 12, 13 are not much informative. Please show only a selection of the most representative case. An additional option is to move these figures as supplementary material.*

**Suggestion accepted. Results shown in figs. 7-8-10-11-12-13 are now shown for representative selected cases in figures 7-9-10.**

---

## Author Response (AR1)

Dear Editor,

We are grateful to the three anonymous referees for their constructive review, encouraging comments and useful suggestions that helped us to improve the quality of the manuscript. Following their comments, we improved the manuscript. In the following lines please find our replies (AR) **in bold fonts** to the reviewers' comments (RC), reported in *italic*. Changes to the manuscript are quoted and reported in **"*bold italic*"** font. Finally, as a first general remark suggests by the Referees #2 and #3, a careful proofreading for English language was performed on the revised version of the manuscript from a professional native English speaker. Here, you can find the revisited text also in comments to Referees.

Kind regards,

Vincenzo Totaro

Andrea Gioia

Vito Iacobellis

Referee #1

*RC. The first is about the assumptions made in this work as their significance with respect to natural phenomena; e.g. the non-stationary model accounts only for the variability of mean in time (which should be explicitly shown for the sake of clarity by reporting the theoretical expressions of the three first order moments a functions of the parameters), yet in nature the non-stationary behavior could imply also a variability in terms of the second order moment. Further, natural time series often depict dependence in time, which significantly affects the power of statistical methods for non-stationarity detection, as also recognized by the Authors themselves. I generally suggest the Authors to improve the discussion on practical limitations of those tests and of the conditions analyzed in their work, yet this is only a personal suggestion to improve the completeness of the discussion.*

**AC. Authors wish to thank the reviewer for these useful suggestions. We introduced in section 2.4 the theoretical expressions by Muraleedharan et al. (2010) of the three first order moments as functions of the parameters. We enlarged the discussion regarding general limits of statistical methods for detection of non-stationarity in both the introduction and the conclusive sections. In particular we specify, also dealing with the final remark from Referee #2, that our purpose here is to show that, in some cases, even a weak linear trend in the mean suffices to reduce power to unacceptable values, then, we decided to limit the investigation to variability of mean in time. Nevertheless, when dealing with natural series, a number of potential other sources of uncertainty, including variability in time of the second order moment, should be considered as we have remarked in the conclusions of the revised manuscript.**

**As main changes to the manuscript, in the introduction the following lines were introduced:**

**"T*he use of null hypothesis significance tests for trend detection has raised concerns and severe criticisms in a wide range of scientific fields since many years (e.g. Cohen, 1994), as outlined by Vogel et al. (2013). Serinaldi et al. (2018) provided an extensive critical review focusing on logical flaws and misinterpretations often related to their misuse.*"**

**and later,**

**"*Nevertheless, as claimed by different authors (Milly et al. 2015, Beven, 2016, among others) the importance of power in earth system sciences fields has been largely overlooked in years while a strong attention is always given to the level of significance (i.e. type I error). As pointed out by Vogel et al. (2013) "a type II error in the context of an infrastructure decision implies under-preparedness, which is often an error much more costly to society than the type I error (overpreparedness)".***

**For changes to conclusions please see response to final remark from Referee #2.**

*RC. Second, I would like to see a deeper comparison with previous literature works on the same topic; e.g. the Authors mention in the conclusion section the paper from Serinaldi et al. (2018), without giving further details. I believe that the comparison with previous literature results could help strengthen the general discussion presented in the conclusion section*

*AC.* **We accepted this suggestion, we decided to move the comparison with previous literature in the introduction. This includes a more specific reference to the work from Serinaldi et al (2018) and also a more general discussion about the use of statistical power for trend detection in hydrology (see response to point 5) of Referee #3 for changes to manuscript).**

**Moreover, in the conclusion section, we introduced some final remarks about the lack of ergodicity to be considered when dealing with nonstationary stochastic process (see response to final remark of Referee #2 for changes to manuscript).**

*RC. Finally, the Authors should spend some efforts to improve the readability of the figures, e.g. by making the lines thick, by increasing the character size etc.*

**AC. Suggestion accepted.**

Referee #2

*RC. The title of the manuscript is not informative of its content. I suggest "Monte Carlo investigation of the power of parametric and non-parametric tests for trend detection in annual maxima series", or something similar.*

**AC. We thank the reviewer for this comment and for the suggested title we slightly changed it "Numerical investigation on the power of parametric and non-parametric tests for trend detection in annual maxima series"**

*RC. Lines 29, 147 and other parts of the manuscript: The statement attributed to Salas (1993) (see line 29) is theoretically incorrect. Stationarity is an attribute of stochastic processes (i.e. models) not of time-series (i.e. their realizations). More precisely, a stochastic process is said to be stationary, if and only if:*

$$X_t \stackrel{\mathrm{d}}{=} X_{t+\tau}, \ \forall \ t, \tau$$

*where $\stackrel{\mathrm{d}}{=}$ denotes equality in all finite-dimensional CDF's $F_{X,n}$, n = 1, 2,.... For example, $F_{X,1}(x;\ t) = F_{X,1}(x)$, $F_{X,2}(x_1, x_2;\ t_1, t_2) = F_{X,2}(x_1, x_2;\ |t_2\text{-}t_1|)$, $F_{X,3}(x_1, x_2, x_3;\ t_1, t_2, t_3) = F_{X,3}(x_1, x_2, x_3;\ |t_2\text{-}t_1|, |t_3\text{-}t_1|)$ and so on.*

*In the above context: a) lack of trends shifts and periodicities in a timeseries does not necessarily mean that the parent process is stationary, b) the wording in the manuscript should be properly modified to avoid use of the terms: "stationary timeseries" and "non-stationary timeseries"*

**AC. We thank the reviewer for this important comment. The statement which is recognized as "theoretically incorrect" is a literal quote from Chapter 18 of Handbook of Hydrology (Maidment, 1993). As mentioned by Koutsoyiannis and Montanari (2015), in their paper section "Semantic and historical review" about the concept of stationarity, this is not the only case. Also the Kendall and Stuart's book make reference to "stationary series", for not counting the large number of papers that may arise from such a keyword search in a database such as Scopus. While we recognize the huge importance of semantic consistency in the scientific literature, we observe that in our paper there is little chance for misconception if considering that we are working in the framework of a Monte Carlo experiment by using time series generated from theoretical models. Nevertheless, in order to avoid any possible confusion, we accepted the reviewer's suggestion by removing the quoted sentence in the introduction (line 29) and introducing a more formal definition of stationarity. Then, we have checked throughout the manuscript for the use of "stationary (and nonstationary) time series". We found, besides the line 147 that was indicated by the reviewer, only one other "suspicious case" at line 264. In lines 147 we rephrased "if the time series is non-stationary, [...]. Vice versa if the time series is stationary" into "*if the time series arises from a non-stationary process, [...]. Vice versa if the process is stationary*"; Line 264 "the generation of stationary series ..." into "*the generation of series from a stationary model...*".**

*RC. The results presented in Figures 7-13 need to be discussed in more detail.*

**AC. We have introduced some more description of such results, nevertheless, following comment 15) from Referee #3, we have also reduced the number of figures and subplots, limiting their display to representative selected cases.**

**The following comment was added to the revised manuscript**

*"Subplots show that the presence of a strong trend coefficient may produce significant loss in the estimator efficiency probably due to deviation from normal distribution of the sample estimates also for long samples. This suggests the need of more robust estimation procedures which provides higher efficiency for estimates of $\varepsilon$ and $\sigma$ in case of strong observed trend. It should be highlighted that efficiency in parameter estimation increases with sample size for $\varepsilon = [0, 0.4]$, while it decreases for both $\varepsilon$ and $\sigma$, in the case $\varepsilon = [-0.4]$, where the trend of the location parameter implies a shift in time of the distribution upper bound."*

**See response to comment 15) from Referee #3 regarding the reduction of figures and subplots.**

*RC. As a final remark, I think that in the concluding section, the Authors should at least comment on an important aspect related to the presented analysis: When inferring the properties of a stochastic process from data, one needs to analyze the available time-series assuming ergodicity.*

*Since a non-stationary process is (by definition) non-ergodic, the stationarity assumption is central to any type of time-series analysis. Hence, non-stationary modeling of physical processes based on data (i.e. a single realization of a stochastic process) is theoretically inconsistent. That said, I believe that the findings of the Authors regarding uncertainty aspects of parametric and non-parametric tests in detecting non-stationarities, significantly underestimate those emerging when real world data is used.*

**AC. Also this comment is particularly welcome because it provide us with the possibility to share our general perspectives on issues concerning real data analysis. Ergodicity, in fact, is not only an important theoretical property of stationary stochastic processes but it also affects practical inferential tasks. Then, we added a final remark about different sources of uncertainty and perspectives about data usage and exogenous information exploitation to be used in environmental change modeling.**

**The following lines were added in conclusions:**

*"As a final remark, concerning real data analysis, in our numerical experiment we showed that, in some cases, a weak linear trend in the mean suffices to reduce power to unacceptable values. Yet we explored the simplest nonstationary working hypothesis by introducing a deterministic linear dependence on time of the location parameter of the parent distribution. Obviously, when making inference from real observed data other sources of uncertainty may affect statistical inference (trend, heteroscedasticity, persistence, nonlinearity, etc), and moreover, if considering a nonstationary process with underlying deterministic dynamics, the process turns out to be non-ergodic, implying that statistic inference from sampled series is not representative of the process's ensemble properties (Koutsoyiannis and Montanari, 2015).*

*As a consequence, while considering a nonstationary stochastic process as produced by a combination of a deterministic function and a stationary stochastic process, other sources of information and deductive arguments should be exploited in order to identify the physical mechanism underlying such relationships. Even in such a case observed time series have a crucial role in order to calibrate and validate deterministic modeling or, in other words, for confirming or disproving the model hypotheses.*

*In the field of frequency analysis of extreme hydrological events, considering the high spatial variability of sample length, trend coefficient, scale and shape parameters, etc, physically based probability distributions could be further developed and exploited for selection and assessment of the parent distribution in the context of non-stationarity and change detection. Physically based probability distributions we refer to are: (i) those arising from stochastic compound processes introduced by Todorovic and Zelenhasic (1970), which include also the GEV (see Madsen et al., 1997) and the TCEV (Rossi et al., 1984), and (ii) the theoretically derived distributions following Eagleson (1972) whose parameters are provided by clear physical meaning and are usually estimated with support of exogenous information in regional methods (e.g. Gioia et al., 2008; Iacobellis et al., 2011; see also for a more extensive overview Rosbjerg et al., 2013).*

*Hence, we believe that "learning from data" (Sivapalan, 2003), will remain in future years a key task for hydrologists facing the challenge of consistently identifying both deterministic and stochastic components of change (Montanari et al., 2013). This involves crucial and interdisciplinary research to develop suitable methodological frameworks for enhancing physical knowledge and data exploitation, in order to reduce the overall uncertainty of prediction in a changing environment."*

Referee #3

Specific comments will be addressed in the following lines according to the same numbering provided by the reviewer:

*RC 1. Lines 113-116. Description of Sen slope estimation and equation (2) should be revised: If N is the number of univocal (non-repeated) couples and j is an index for the j-th couple ($x_i$, $x_k$), why should be j > k? Maybe the authors mean i > k? Please check and better specify the role of j index. Remove also "Sorting in ascending order ....", declaring that the median values is the final estimate is enough and the reader understand.*
**AC. Suggestion accepted.**

*RC 2. Lines 179-188 + Appendix. These lines + Appendix should be removed. All the analyses in the manuscript are based on ML estimates, thus there is no reason to keep a description of PWM and L-moments.*
**AC. Suggestion accepted, the appendix and the expressions of L-moments were removed and replaced by theoretical expressions of moments as from first comment from Referee #1.**

*RC 3. Section 2.5. It should be written that stationarity is assumed as null hypothesis (e.g. in line 207 and 210)*
**AC. Suggestion accepted, we modified lines 205 and 211 by adding, respectively, that "*the null hypothesis of stationarity is false*" and "*the null hypothesis of stationarity (which) is true*".**

*RC 4. Section 2.5, lines 201 and 209. Just a curiosity: why experiments are conducted with a different number of samples (2000 vs 10000)?*
**AC. We used N = 10000 generations to be sure that the effect of sample variability is negligible when estimating the real level of significance and the corresponding threshold of acceptance. For the evaluation of power (i.e. in all generations with values of $\zeta_1 \neq 0$) we found N = 2000 is a good compromise between quality of results and reasonable computational time. To this purpose we performed sample checks with N = 10000 obtaining negligible differences. We did not report such details for the sake of simplicity and also because power values obtained with generations with $\zeta_1 \neq 0$ are only shown in graphical form and as such they are not even distinguishable from those shown. Moreover, N = 2000 is the same number of generations used by Yue et al. (2002a). In the revised manuscript we added this short explanation about this point in section 2.5:**
**"*We used a reduced number of generations (N = 2000) for the evaluation of power as good compromise between quality of results and computational time and, also in analogy with Yue et al. (2002a).*"**
**Regarding the number of generations N, see also answers to points 7), 8) and 14).**

*RC 5. Section 2.5, line 198. Authors can provide a reference for the sentence "1-to-4 trade-off between α and β is accepted"?*

**AC. Such question is everything but trivial, we can provide a reference about this (Cohen, 1994) and we introduced it in the revised manuscript, nevertheless such paper does not come from hydrological literature but from Psychology and is by far the most cited paper in Scopus about "statistical power". Reflecting about this point has stimulated a discussion about the apparent lack of references of this type in the earth system sciences that we have added in the conclusion section. The following lines were added in the conclusion section:**

*"Considering the feasibility of numerical evaluation of power, allowed by the parametric approach, we observe that, while the awareness of the crucial role of type II error is growing in latest years in the hydrological literature, a common debate would deserve more development about which power values should be considered acceptable. Such an issue is much more enhanced in other scientific fields where the experimental design is traditionally required to estimate the appropriate sample size to adequately support results and conclusions. In psychological research, Cohen (1994) proposed 0.8 as a conventional value of power, to be used with level of significance 0.05 thus leading to a ratio 4 to 1 between the risk of type II and type 1 error. The conventional value proposed by Cohen (1994) has been taken as reference by thousands of papers in social and behavioural sciences. In pharmacological and medical research, depending on real implications and nature of the type II error, conventional values of power may be as high as 0.999. This is the value suggested by Liebher (1990), when testing a treatment for patients' blood pressure. He stated, while "guarding against cookbook application of statistical methods", that "it should also be noted that, at times, type II error may be more important to an investigator then type I error".*

*We believe that, selecting between stationarity and non-stationarity models for extreme hydrological event prediction, a fair comparison between the null and the alternative hypotheses as $\alpha = \beta = 0.05$ should be taken, which provides power = 0.95. In our discussion we considered 0.8 as minimum threshold for acceptable power values."*

*RC 6. Lines 214-219. I understand the choice of GEV parameters and it is reasonable to my knowledge of rainfall maxima in Mediterranean climate. I was wandering whether it can be more informative to present results and figures in a more general way, e.g. as a function of the relative trend $\zeta_1/\zeta_0$ (which has dimension 1/time)?*

**AC. This comment is particularly important and stimulating. Nevertheless, such generalization is out of the purpose of this paper and would require more extensive investigation. In facts, presenting results in terms of the ratio $\zeta_1/\zeta_0$ would make sense if results of the analyses were the same for different couples of $\zeta_1$ and $\zeta_0$ values producing the same ratio. Actually, this is not the case when $\sigma$ is fixed. We believe that invariant properties of the frequency distribution of rainfall or floods annual maximum values could be exploited for a generalization of these analyses, but this would involve consideration of scaling features of different order moments. This could be a quite interesting future development of this study, involving also time dependence of the scale parameter. Not any change has been done to the manuscript.**

*RC 7. Line 236. "a multi-peak . . ..": are you sure that it is not a sampling effect? Increasing the number of simulations is the result the same?*

**AC. In this figure we show some distributions of sampled errors in terms of difference between the power obtained with AIC and the power obtained with AIC$_c$. The curves show that the entire range of sampled errors provides negligible values compared to the expected power values. The different peaks in one curve (L = 30) are observed because we merged sample errors obtained for different values of $\sigma$ characterized by a small different (random) bias. We added such considerations in the revised paper:**

*"Aim of this figure is to show that the difference between the power obtained with AIC and the power obtained with AIC$_c$ is negligible. Anyway, different peaks in one curve (L = 30) can be explained by merging sample errors obtained for different values of σ."*

*RC 8. Lines 267-273 and Table 1. Similar to previous comment: are you sure that variability observed for different σ (but keeping constant the other constraints) are not a sampling effect? Increasing the number of simulations is the result the same?*

**AC. Results shown in Table 1 are, to some extent, affected by a sampling effect. We had numerically checked the sample variability of the actual level of significance, which is quite smaller than the difference between the designed (0.05) and the actual value of the significance level for the LR test. We didn't report such analysis for the sake of simplicity nevertheless we used a very high number of generations (N = 10000) to produce these values (see also response to comment #4). On the other hand, we agree that there is not such evidence with specific reference to different σ values, while ε and L mostly affect results and accordingly we revised the manuscript rephrasing the sentence as:**

*"Such effect is exalted when the parent distribution is upper bounded (ε=-0.4) and for shorter series (L = 30)."*

*RC 9. Line 301. I would specify here that series are stationary.*

**AC. Suggestion accepted**

*RC 10. Section 3.4 (and maybe other parts of the manuscript, figures and tables). GEV parameters are always estimated by ML, thus I suggest to avoid the use of the prefix "ML-" before the symbol of the parameter (e.g. in Line 310 and 313). This would made more clear text, figure and tables.*

**AC. We would rather maintain the use of the prefix "ML-" in order to distinguish between values (ML-ε, ML-σ, ML-ζ$_1$) estimated from the series and the theoretical values (ε, σ, ζ$_1$) used in the parent distribution. Not any change was done.**

*RC 11. Line 316. "Figs. 6 and 9": usually figures should be ordered as they are cited.*

**AC. Suggestion accepted by revising the text by removing such reference to fig. 9 which is introduced later.**

*RC 12. Figures. Please consider the opportunity to use larger fonts for labels, they are not readable here, and most probably Figure will be reduced in the final formatting.*

**AC. Suggestion accepted**

*RC 13. Figure 6. Use the same range of scales (e.g. 0-0.6) in the y-axis for a fair comparison.*
**AC. Suggestion accepted**

*RC 14. Figure 9. Again: are you sure that fluctuations are not due to sampling effects? See e.g. the subplot in the right part of the Figure 9.*
**AC. We checked such results with different sets of random generations also increasing N up to 10000. Qualitatively results do not change. "*Randomness of results for $L = 30$ and $\sigma = 15, 20$ is probably due to a reduced efficiency of the algorithm that maximizes the log-Likelihood function, for heavy tailed distributions.*" Such consideration was added in the revised manuscript (Fig. 9 is now Fig. 8, because of the following comment).**

*RC 15. Figures 7, 8, 10, 11, 12, 13 are not much informative. Please show only a selection of the most representative case. An additional option is to move these figures as supplementary material.*
**AC. Suggestion accepted. Results shown in figs. 7-8-10-11-12-13 are now shown for representative selected cases in figures 7-9-10.**

**Authors' supplement to response at comments from Referees: references.**

Following the general comment and suggestions from Referee #1, involving also the final remark of Referee #2, and comment 5) from Referee # 3, asking for more insights about the implications of this work within the general framework of real data analysis, we have introduced in the revised manuscript a number of references hereafter reported with indication of the position in the revised manuscript.

**Section 1 Introduction, see response to 1st comment from Referee #1:**

**Beven, K.**: Facets of uncertainty: Epistemic uncertainty, non-stationarity, likelihood, hypothesis testing, and communication, Hydrol. Sci. J., doi:10.1080/02626667.2015.1031761, 2016.

**Cohen, J.**: The earth is round (p < .05), American Psychol., 49, 997–1003, 1994.

**Milly, P. C. D.,** Betancourt, J., Falkenmark, M., Hirsch, R. M., Kundzewicz, Z. W., Lettenmaier, D. P., Stouffer, R. J., Dettinger, M. D. and Krysanova, V.: On Critiques of "stationarity is Dead: Whither Water Management?" Water Resour. Res., doi:10.1002/2015WR017408, 2015.

**Vogel, R. M.,** Rosner, A. and Kirshen, P. H.: Brief communication: Likelihood of societal preparedness for global change: Trend detection, Nat. Hazards Earth Syst. Sci., doi:10.5194/nhess-13-1773-2013, 2013.

**Section 2.4 The GEV parent distribution, see response to 1st comment from Referee #1:**

**Muraleedharan, G.,** Guedes Soares, C. and Lucas, C.: Characteristic and moment generating functions of generalised extreme value distribution (GEV), in Sea Level Rise, Coastal Engineering, Shorelines and Tides., 2011.

**Section 4 Conclusions, see response to 2nd comment from Referee #1 and final remark from Referee #2:**

**Eagleson, P. S**.: Dynamics of flood frequency, Water Resour. Res., doi:10.1029/WR008i004p00878, 1972.

**Gioia, A**., Iacobellis, V., Manfreda, S. and Fiorentino, M.: Runoff thresholds in derived flood frequency distributions, Hydrol. Earth Syst. Sci., doi:10.5194/hess-12-1295-2008, 2008.

**Iacobellis, V**., Gioia, A., Manfreda, S. and Fiorentino, M.: Flood quantiles estimation based on theoretically derived distributions: Regional analysis in Southern Italy, Nat. Hazards Earth Syst. Sci., doi:10.5194/nhess-11-673-2011, 2011.

**Madsen, H**., Rasmussen, P. and Rosbjerg, D.: Comparison of annual maximum series and partial duration series for modelling exteme hydrological events: 1. At sit modelling, Water Res. Res., 1997.

**Montanari, A.,** Young, G., Savenije, H.H.G., Hughes, D., Wagener, T., Ren, L.L., Koutsoyiannis, D., Cudennec, C., Toth, E., Grimaldi, S., Blöschl, G., Sivapalan, M., Beven, K., Gupta, H., Hipsey, M., Schaefli, B., Arheimer, B., Boegh, E., Schymanski, S.J., Di Baldassarre, G., Yu, B., Hubert, P.,

Huang, Y., Schumann, A., Post, D., Srinivasan, V., Harman, C., Thompson, S., Rogger, M., Viglione, A., McMillan, H., Characklis, G., Pang, Z., and Belyaev, V., "Panta Rhei—Everything Flows": Change in hydrology and society—The IAHS Scientific Decade 2013–2022. Hydrological Sciences Journal. 58 (6) 1256–1275, 2013.

**Rosbjerg, D.**, Blöschl, G., Burn, D., Castellarin, A., Croke, B., Di Baldassarre, G., V. Iacobellis, T. R. Kjeldsen, G. Kuczera, R. Merz, A. Montanari, D. Morris, T. B. M. J. Ouarda, L. Ren, M. Rogger, J. L. Salinas, E. Toth, and Viglione, A.: Prediction of floods in ungauged basins. In G. Blöschl, M. Sivapalan, T. Wagener, A. Viglione, & H. Savenije (Eds.), Runoff Prediction in Ungauged Basins: Synthesis across Processes, Places and Scales (pp. 189-226). Cambridge: Cambridge University Press. doi:10.1017/CBO9781139235761.012, 2013.

**Rossi, F.**, Fiorentino, M. and Versace, P.: Two-Component Extreme Value Distribution for Flood Frequency Analysis, Water Resour. Res., doi:10.1029/WR020i007p00847, 1984.

**Sivapalan, M.**, Prediction in Ungauged Basins: A Grand Challenge for Theoretical Hydrology, Hydrol. Process. 17, 3163–3170, 2003.

**Todorovic, P.** and Zelenhasic, E.: A Stochastic Model for Flood Analysis, Water Resour. Res., doi:10.1029/WR006i006p01641, 1970.

**Section 4 Conclusions, see response to 2nd comment from Referee #1 and comment 5) from Referee #3:**

**Cohen, J.**, A power primer, Psychological Bulletin, Vol 112(1), Jul 1992, 155-159, 1992.

**Lieber, R. L.**: Statistical significance and statistical power in hypothesis testing, J. Orthop. Res., doi:10.1002/jor.1100080221, 1990.

Dear Editor

please, find below a list of all relevant changes to the original version of the manuscript. For the sake of clarity, we report rows numbers of the original manuscript, stating the corresponding comment of Referee which justified modification. We use an ***Italic bold*** font when reporting changes in the manuscript.

Sincerely

Vincenzo Totaro

Andrea Gioia

Vito Iacobellis

List of relevant changes to submitted manuscript:

- Lines 1-2: accepting suggestion n. 1 of Referee #2, we changed title of the manuscript in "***Numerical investigation on the power of parametric and non-parametric tests for trend detection in annual maximum series***".

- Line 27: some short historical notes on the concept of stationarity is introduced (suggestion n. 2 of Referee #2).
  ***Kolmogorov in 1931 introduced the concept of stationarity of a probability distribution, formally defined by Khintchine in 1934, as depicted in the historical review provided by Koutsoyiannis and Montanari (2015)***

- Line 48: in order to improve the part about practical limitations of tests used into a non-stationary framework, we introduced a discussion about the mutual importance of type I and type II errors in statistical tests (see comment n.1 of Referee #1).
  ***On the other hand, the use of null hypothesis significance tests for trend detection has raised concerns and severe criticisms in a wide range of scientific fields since many years (e.g. Cohen, 1994), as outlined by Vogel et al. (2013). Serinaldi et al. (2018) provided an extensive critical review focusing on logical flaws and misinterpretations often related to their misuse.***
  ***In general, the use of statistical tests involves different errors, such as type I (rejecting the null hypothesis when it is true) and type II (accepting the null hypothesis when it is false). The latter is related to the test power, i.e. the probability of rejecting the null hypothesis when it is false but, as recognized by a few authors (e.g. Milly et al. 2015; Beven, 2016), in years the importance of power has been largely overlooked in earth system science fields. A strong attention has always been given to the level of significance (i.e. type I error) though, as pointed out by Vogel et al. (2013), "a type II error in the context of an infrastructure decision implies under-preparedness, which is often an error much more costly to society than the type I error (over-preparedness)".***
  ***Moreover, as already proven by Yue et al. (2002a), the power of the Mann-Kendall test, despite its non-parametric structure, actually shows a strong dependence on the type and parametrization of the parent distribution***

- Lines 179-187: we replaced the discussion on L-moments with the expression of the three first order moments of GEV distribution, as suggested by comment n.1 of Referee #1.
  ***According to Muraleedharan et al. (2010), the first three moments of GEV distribution are:***

$$Mean = \zeta + \frac{\sigma}{\varepsilon}(g_1 - 1) \qquad\qquad \varepsilon \neq 0, \varepsilon < 1 \qquad\qquad (10)$$

$$Variance = \frac{\sigma^2}{\varepsilon^2}(g_2 - g_1^2) \qquad\qquad \varepsilon \neq 0, \varepsilon < \frac{1}{2} \qquad\qquad (11)$$

$$Skewness = sgn(\varepsilon) \cdot \frac{g_3 - 3g_2g_1 + 2g_1^3}{(g_2 - g_1^2)^{3/2}} \qquad\qquad \varepsilon \neq 0, \varepsilon < \frac{1}{3} \qquad\qquad (12)$$

*where $g_k = \Gamma(1 - k\varepsilon)$, with $k \in \mathbb{Z}^+$ and $\Gamma(\cdot)$ is the Gamma function. It is worth noting that, following Eqs. (10), (11) and (12), the trend in the position parameter only affect the Mean while Variance and Skewness remain constant.*

− Line 198: we added a sentence for better specifying values of tests' level of significance and power we consider acceptable for our purposes.

*Then, in our experiment we assumed always significance level 0.05, and, for the following description of results and discussion we considered a power level less than 0.8 as too low and hence unacceptable. In the conclusions section we report further considerations about this choice.*

− Line 212: We justified the use of the number N = 2000 as selected number of simulations (in the light of comment n.4 of Referee #3).

*We used a reduced number of generations (N = 2000) for the evaluation of power as good compromise between quality of results and computational time and, also in analogy with Yue et al. (2002a).*

− Line 237: We added a sentence (according to comment n.7 of Referee #3) for commenting the use of Fig. 2.

*The purpose of this figure is to show that the difference between the power obtained with AIC and the power obtained with $AIC_c$ is negligible. Different peaks in one curve (L = 30) can be explained by the merge of sample errors obtained for different values of σ.*

− Line 327: we added a sentence (according to comment n.14 of Referee #3) for interpreting of fluctuations in Fig. 9 (which become Fig. 8 in revised manuscript).

*Randomness of results for $L = 30$ and $\sigma = [15, 20]$ is probably due to a reduced efficiency of the algorithm that maximizes the log-Likelihood function, for heavy tailed distributions.*

− Line 334: Further comments to figures are introduced (according to point n.3 of Referee #2).

*It should be highlighted that efficiency in parameter estimation increases with sample size for $\varepsilon = [0, 0.4]$, while it decreases for both ε and σ, in the case $\varepsilon = [-0.4]$, where the trend of the location parameter implies a shift in time of the distribution upper bound.*

− Line 338: Moving from comment n. 5 of Referee #3, we decided to introduce a discussion about the importance of dealing with type II error in statistical test used in Earth sciences.

[revised manuscript text omitted]

− Lines 396-415: Accepting suggestions n.1 of Referee #1 and n. 2 of referee #3, we removed Appendix A.

− References: we introduced new references added in the modified version of manuscript.

− All figures: accepting suggestions of Referees #1 and #3, we improved readability of all figures.

− Figures 7-8: we condensed Figures 7-8 into a single Fig. 7 in the final version of the manuscript.

− Fig. 9 become Fig. 8:

− Figures 10-11: we condensed Figures 10-11 into a single Fig. 9 in the final version of the manuscript.

− Figures 12-13: we condensed Figures 12-13 into a single Fig. 10 in the final version of the manuscript.

Please, note that changes of Figures were made for fulfilling comments n. 3 of Referee #2 and n. 15 of Referee #3.

[revised manuscript text omitted]